# *MAP1B* mutations cause intellectual disability and extensive white matter deficit

G. Bragi Walters[1,2], Omar Gustafsson[1], Gardar Sveinbjornsson[1], Valgerdur K. Eiriksdottir [1],
Arna B. Agustsdottir[1], Gudrun A. Jonsdottir[1], Stacy Steinberg [1], Arni F. Gunnarsson[1], Magnus I. Magnusson[1],
Unnur Unnsteinsdottir[1], Amy L. Lee[1], Adalbjorg Jonasdottir[1], Asgeir Sigurdsson[1], Aslaug Jonasdottir[1],
Astros Skuladottir[1], Lina Jonsson[1,3], Muhammad S. Nawaz [1,2], Patrick Sulem [1], Mike Frigge[1], Andres Ingason[1],
Askell Love[2,4], Gudmundur L. Norddhal[1], Mark Zervas[1], Daniel F. Gudbjartsson [1,5], Magnus O. Ulfarsson [1,6],
Evald Saemundsen [2,7], Hreinn Stefansson [1] & Kari Stefansson[1,2]

Discovery of coding variants in genes that confer risk of neurodevelopmental disorders is an important step towards understanding the pathophysiology of these disorders. Whole-genome sequencing of 31,463 Icelanders uncovers a frameshift variant (E712KfsTer10) in *microtubule-associated protein 1B* (*MAP1B*) that associates with ID/low IQ in a large pedigree (genome-wide corrected $P = 0.022$). Additional stop-gain variants in *MAP1B* (E1032Ter and R1664Ter) validate the association with ID and IQ. Carriers have 24% less white matter (WM) volume ($\beta = -2.1$SD, $P = 5.1 \times 10^{-8}$), 47% less corpus callosum (CC) volume ($\beta = -2.4$SD, $P = 5.5 \times 10^{-10}$) and lower brain-wide fractional anisotropy ($P = 6.7 \times 10^{-4}$). In summary, we show that loss of MAP1B function affects general cognitive ability through a profound, brain-wide WM deficit with likely disordered or compromised axons.

[1] deCODE genetics/Amgen, Reykjavik 101, Iceland. [2] Faculty of Medicine, University of Iceland, Reykjavik 101, Iceland. [3] Department of Pharmacology, Institute of Neuroscience and Physiology, Sahlgrenska Academy, University of Gothenburg, Gothenburg 405 30, Sweden. [4] Department of Radiology, Landspitali University Hospital, Fossvogur, Reykjavik 108, Iceland. [5] School of Engineering and Natural Sciences, University of Iceland, Reykjavik 101, Iceland. [6] Faculty of Electrical and Computer Engineering, University of Iceland, Reykjavik 101, Iceland. [7] The State Diagnostic and Counselling Centre, Kopavogur 200, Iceland. Correspondence and requests for materials should be addressed to H.S. (email: hreinn@decode.is) or to K.S. (email: kstefans@decode.is)

ID refers to a group of aetiologically diverse conditions, resulting in below average cognitive abilities. Affected individuals are typically identified in childhood due to delay in developmental milestones. A formal diagnosis of ID is made when an intelligence quotient (IQ) of less than 70, on a normal distribution with a mean of 100 and standard deviation of 15, is coupled with impairments in adaptive function including conceptual, social and practical skills (APA URL)[1]. A full-scale IQ (FSIQ) score is calculated from subtests grouped into two general areas, verbal and performance related tasks, which can be discrepant in some disorders[2,3]. While ID negatively affects fecundity, de novo mutations maintain ID sequence variants in the population[4] and heritability estimates are higher for mild than for severe ID[5].

Numerous genes have been implicated across a wide spectrum of syndromic and non-syndromic forms of ID and a significant overlap noted with genes identified in other neurodevelopmental disorders including ASD[6]. Large-scale next-generation sequencing efforts of individuals with developmental disorders[7], ID[8,9] and ASD[10] have made significant contributions to the understanding of their pathogenesis. Bioinformatics and functional analyses have grouped some of the ID genes into common pathways, representing regulation of transcription and translation, neurogenesis and neuronal migration and synaptic function and plasticity[11].

Utilising next-generation sequencing to search for variants in genes important for brain development and function combined with psychological assessment and brain structural traits in carriers, could provide a better understanding of the molecular processes and neuronal connectivity underlying cognitive diversity in neurodevelopmental disorders. Here we describe three variants in the *MAP1B* gene, which truncate the protein, segregating with ID and low IQ. Furthermore, carriers have significantly less white matter (WM) volume and a decrease in fractional anisotropy (FA), suggesting disordered or compromised axons.

## Results

**Genome-wide scan for high-impact variants.** We recruited an extended pedigree with ID for cognitive phenotyping, brain imaging and whole-genome sequencing (WGS) (Fig. 1a and Supplementary Table 1). In this family there are five individuals with diagnosed ID and three of their relatives, without a clinical diagnosis, with IQ below 70. We applied a recently described statistical procedure[12] to estimate the intra-family association, of genome-wide, rare, coding variants, with diagnosed ID or IQ below 70. Only one variant, a frameshift variant causing premature truncation (E712KfsTer10) of *MAP1B* (Supplementary Figure 1) was found to associate with the cognitive impairments in this family (genome-wide corrected $P = 0.022$).

The E712KfsTer10 variant is carried by eight members of this family, seven with ID or IQ below 70, and not found in any of the other 31,463 WGS Icelanders. The eighth carrier (FAM1-A1) is deceased, and so cognitive assessment could not be performed. We note a noncarrier, in this family, with a diagnosis of ID (FAM1-D4). As the E712KfsTer10 variant results in cognitive impairments a carrier is likely to find a partner with comparable cognitive abilities. This may introduce, into the family, a mixture of rare and common markers conferring risk of cognitive deficits.

**MAP1B association with ID confirmed.** We searched for additional *MAP1B* LoF variants in the 31,463 WGS Icelanders, of whom 408 carry the diagnosis of ID, and found two stop-gain variants, E1032Ter and R1664Ter (Supplementary Figure 1). The E1032Ter variant was found in four individuals: three siblings and their mother (Fig. 1b and Supplementary Table 1). Two of the three siblings have atypical autism and a FSIQ below 70. Both have higher verbal IQ (VIQ) than performance IQ (PIQ) and a cognitive profile consistent with non-verbal learning disorder (NVLD)[13]. The third sibling with a FSIQ of 79, reported having dyscalculia and dyslexia, while their mother has a FSIQ of 77

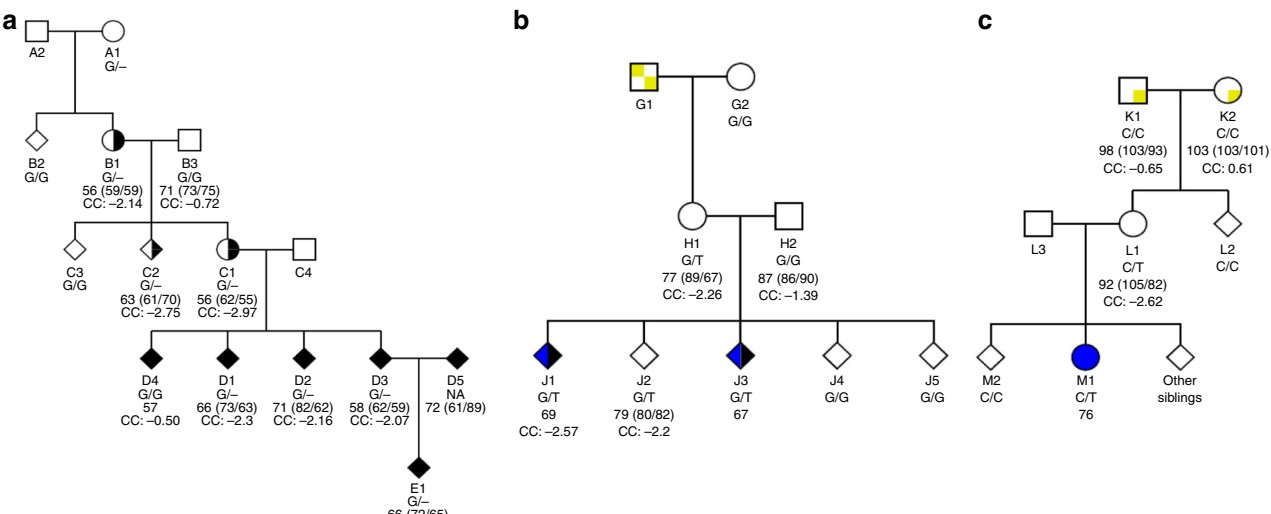

**Fig. 1** Pedigree plots for *MAP1B* LoF carrier families. **a** Family 1 (FAM1)—E712KfsTer10 (c.2133delG), **b** Family 2 (FAM2)—E1032Ter (c.3094 G > T) and **c** Family 3 (FAM3)—R1664Ter (c.4990 C > T). Squares are male, circles are female, diamonds are where gender is withheld, unfilled are unaffected, filled with black are intellectual disability affected, half black and white are individuals without a clinical diagnosis but an IQ below 70, blue are Autism spectrum disorder affected and yellow and white squares within the pedigree symbol indicates where it is most likely the initial *MAP1B* LoF mutation event occurred. Below each icon is the subjects: alias; *MAP1B* LoF variant genotype; Full-scale IQ (Verbal IQ/Performance IQ) from WASI[IS] or WPPSI-R (FAM1-B1,-B3,-C1,-C2,-D1,-D2,-D3; FAM2-H1,-H2,-J2; FAM3-K1,-K2,-L1 or FAM1-D5,-E1, respectively), or only Full-scale IQ reported from WISC-IV (FAM1-D4, FAM2-J1,-J3 and FAM3-M1), and corpus callosum (CC) volume (all individuals with structural MRI also have DTI except FAM1-C1). Refer to Supplementary Figure 2 for a description of how the de novo event carriers were identified

(VIQ higher than PIQ). The R1664Ter variant was found in two individuals, a mother with a FSIQ of 92 (VIQ 105, PIQ 82) and her daughter with a diagnosis of Asperger syndrome and a FSIQ of 76 (Fig. 1c and Supplementary Table 1).

Hence, together the E1032Ter and R1664Ter variants are carried by two ID cases and four of their relatives without an ID diagnosis, but have an average IQ score of 81. Regressing the allele carrier status, for either LoF variant, on the ID status or IQ scores, further confirmed the association with LoF variants in *MAP1B* (OR = 69.6, $P = 4.9 \times 10^{-4}$ and Effect (IQ score) = −23.07, $P = 1.4 \times 10^{-3}$, respectively; Methods). This establishes that LoF variants in *MAP1B* associate strongly with cognitive impairments.

We also evaluated anthropometric traits including head circumference, height and weight and found no association with *MAP1B* LoF carrier status, after adjusting for relatedness ($P = 0.16, 0.98, 0.56$, respectively; Supplementary Table 2).

**MAP1B variants in public databases.** The three *MAP1B* LoF variants, described here, have not been previously reported (Supplementary Table 3) and were not observed in any of the publicly available databases (Supplementary Table 4), which is consistent with their rarity in Iceland. Furthermore, the Exome Aggregation Consortium (ExAc, URL)[14] estimated a very-high probability of LoF intolerance for *MAP1B*, suggesting that LoF mutations in *MAP1B* reduce fitness. However, LoF variants in *MAP1B* are found in publicly available databases at a frequency of 1 in 10,000. ExAc also classify missense variants in *MAP1B* to be under modest constraint. However, we do not observe any *MAP1B* missense variants in individuals with ID or ASD.

Copy number variations spanning *MAP1B* have also been reported in individuals where the most frequently described trait is ID or developmental delay (Supplementary Table 5). A single individual with a deletion spanning *MAP1B* was identified in a sample of 157,065 Icelandic genotyped subjects. This deletion carrier has severe ID (FAM4; Supplementary Table 1).

**MAP1B carriers show general cognitive impairments.** Overall, the *MAP1B* LoF carriers ($n = 13$) have lower FSIQ than non-carriers ($P = 8.2 \times 10^{-6}$, Table 1 and Supplementary Figure 3) and a paired test revealed higher VIQ than PIQ ($P = 0.042$, two-tailed, Student's t-test), mainly due to the lower Block Design subtest. A comparable difference between VIQ and PIQ scores has been reported in disorders that include NVLD like symptoms, including autism spectrum disorder (ASD)[15], Fragile X syndrome[13] and agenesis of corpus callosum (AgCC)[16].

*MAP1B* LoF carriers and noncarrier relatives were administered a battery of neuropsychological tests gauging aspects of executive function, memory, attention and fluency[17]. As expected from the IQ assessment, *MAP1B* LoF carriers performed poorly on all cognitive tasks, in particular they recalled fewer words in the letter fluency test (LF) and performed slower on the trail-making test part A (TMT-A) (Table 2). Notably, *MAP1B* LoF carriers also made more mistakes on the emotion recognition task (ERT-Cor), a test of a form of social cognition, where comparable impairments are described in ASD[2], AgCC[18] and NVLD[13]. The MINI[19] interview did not reveal any psychiatric disorders and overall social and occupational functioning according to the global assessment of functioning scale (GAF)[20] gave comparable scores for carriers and noncarriers ($P = 0.69$) (Table 2).

**IQ and education polygenic score in MAP1B carriers.** Combining the effect sizes from a genome-wide association study (GWAS) to generate a polygenic score (PGS) is beginning to play a role in predicting differences in complex traits such as

intelligence[21] and educational attainment[22]. We used summary statistics from a recent GWAS for IQ[23] and educational attainment[24] to determine the per-locus allele-specific weightings of markers used to calculate a PGS for IQ (PGS-IQ) and educational attainment (PGS-EDU), respectively (Methods and References[22,25]). We compared the PGS-IQ and PGS-EDU in *MAP1B* carriers ($n = 13$) to controls ($n = 2164$) and found no significant difference ($P = 0.20$ and 0.82, respectively; Welch Two Sample t-test) (Supplementary Figure 4 and Supplementary Figure 5).

**Brain structure affected by MAP1B variants.** Brain MRI scans were obtained of ten *MAP1B* LoF carriers. They have a smaller corpus callosum (CC) and less brain-wide WM volume ($\beta = -2.4$ SD, $P = 5.5 \times 10^{-10}$ and $\beta = -2.1$ SD, $P = 5.1 \times 10^{-8}$, respectively) than noncarriers ($n = 949$) (Table 3, Fig. 2 and Supplementary Data 1). This corresponds to a CC and WM volume deficit of 47 and 24%, respectively. Noncarrier relatives ($n = 5$) have mean CC and WM volumes (−0.53 (SD 0.72) and −0.88 (SD 0.97), respectively) indistinguishable from controls ($P = 0.22$ and 0.043, respectively) (Supplementary Figure 6). The association between the *MAP1B* LoF variants and smaller CC and brain-wide WM volume is consistent with published findings in mouse *MAP1B* knockouts, where AgCC, enlargement of the lateral and third ventricles, and abnormal structures of the cortex, cerebellum and hippocampus have been described[26–28].

Apart from the reduced WM, significant differences were observed in the grey matter (GM) surface area and volume of several regions including decrease in right hemisphere (RH) lateral occipital surface area, RH isthmus cingulate, thalamus and left hemisphere (LH) postcentral surface area (Table 4 and Fig. 2). The only region with more GM in carriers than noncarriers is the RH insula (Table 4 and Fig. 2). The increased RH insula thickness may be comparable with the recent description of deep perisylvian/insular polymicrogyria on the right side, in a mother and child, both with a truncating mutation in *MAP1B* (Arg1106Ter), and diagnosed with periventricular nodular heterotopia (PVNH)[29].

**Clinical MRI evaluation of MAP1B variant carriers.** We performed clinical MRI evaluation of the *MAP1B* LoF variant carriers described here and found nine to have periventricular heterotopia symptoms consistent with PVNH (Supplementary Table 1). Furthermore, most of the carriers had unusual/asymmetric supratentorial brain anatomy, without any gross malformation except in one *MAP1B* LoF carrier (FAM1-C1) with isolated inferior vermian hypoplasia (Dandy-Walker variant). A common neurological symptom of PVNH is epilepsy. However, only one *MAP1B* LoF carrier, in our cohort, has diagnosed epilepsy (FAM1-D3).

**FA in reduced WM volume.** Insight into WM microstructure can be deduced from diffusion tensor imaging (DTI)[30]. Applying tract-based spatial statistic (TBSS) analysis to whole-brain diffusion data, from nine *MAP1B* LoF carriers compared to 181 controls, revealed an association with reduced FA in brain-wide WM tracts (Fig. 3a). The lower FA observed in *MAP1B* LoF carriers suggests that their axonal tracts are either of reduced integrity or disordered arrangement. We also evaluated a subset of the noncarrier controls with low CC or WM and found that their FA was only affected in the densest part of the CC (Fig. 3b). This suggests that the effect of the *MAP1B* LoF variants on FA may be more profound than a smaller WM volume would indicate.

**Table 1 IQ scores for *MAP1B* LoF carriers compared with controls**

| IQ | n (MAP1B—Carriers/Controls) | Mean (SD) (MAP1B—Carriers/Controls) | β (95% CI) | P |
|---|---|---|---|---|
| Full-scale IQ | 13/2226 | 68.3 (10.5)/102.1 (14.9) | −1.6 (−2.3, −0.9) | $8.2 \times 10^{-6}$ |
| Performance IQ | 10/2226 | 66.4 (9.3)/99.8 (15.2) | −1.6 (−2.4, −0.8) | $6.4 \times 10^{-5}$ |
| Block design | 9/1768 | 27.6 (5.8)/49.4 (10.5) | −1.7 (−2.5, −0.9) | $3.1 \times 10^{-5}$ |
| Matrix reasoning | 9/1768 | 31.4 (7.5)/49.7 (10.0) | −1.0 (−1.8, −0.2) | 0.014 |
| Verbal IQ | 10/2226 | 74.5 (14.8)/103.5 (15.0) | −1.2 (−2.0, −0.4) | 0.0039 |
| Vocabulary | 9/1767 | 33.2 (11.8)/52.3 (9.5) | −1.1 (−1.9, −0.3) | 0.0082 |
| Similarities | 9/1768 | 34.0 (10.8)/51.5 (9.7) | −1.0 (−1.8, −0.2) | 0.016 |

The IQ tests values are unadjusted means and standard deviation (SD). For analyses, the IQ test scores were inverse normal transformed, then shifted and scaled, resulting in controls having a mean of 0 and a standard deviation of 1. Lower IQ scores represent greater impairment in *MAP1B* LoF carriers. The effects (β in SD) and P-values were calculated by comparing *MAP1B* LoF carriers (FAM1-B1,-C1,-C2,-D1,-D2,-D3, FAM2-H1,-J2, FAM3-L1) with controls using a generalised least-squares regression with a variance–covariance matrix based on the kinship coefficient of each pair of individuals
SD: standard deviation

**Table 2 Cognitive assessment test scores for *MAP1B* LoF carriers compared with controls**

| Cognitive test | n (MAP1B—Carriers/Controls) | Mean (SD) (MAP1B—Carriers/Controls) | β (95% CI) | P |
|---|---|---|---|---|
| LMI&II | 9/2230 | 65.9 (22.5)/68.6 (17.3) | 0.3 (−0.5, 1.1) | 0.45 |
| LF | 9/2221 | 16 (8.5)/29.3 (9.5) | 1.2 (0.4, 2.1) | 0.0037 |
| CF | 9/2224 | 17.2 (5.6)/22.0 (5.5) | 0.8 (−0.01, 1.6) | 0.053 |
| Stroop | 8/2218 | 48.8 (31.5)/26.6 (12.3) | 1.0 (0.2, 1.9) | 0.018 |
| TMT | 8/2207 | 82.8 (75.5)/44.8 (26.2) | 0.8 (−0.1, 1.6) | 0.065 |
| Pers. err | 9/2175 | 24.1 (15.2)/14.7 (12.3) | 0.9 (0.1, 1.8) | 0.031 |
| SWM | 9/2208 | 6.2 (5.1)/6.3 (7.2) | 0.9 (0.1, 1.8) | 0.026 |
| RVIP | 7/1964 | 0.9 (0.05)/0.9 (0.1) | 0.9 (0.1, 1.8) | 0.029 |
| TMT-A | 8/2218 | 42.2 (15.8)/29.6 (12.8) | 1.2 (0.4, 2.1) | 0.0046 |
| Str-bl | 8/2223 | 22.9 (4.1)/21.3 (4.4) | 0.2 (−0.7, 1.0) | 0.72 |
| ERT-Cor | 6/788 | 0.5 (0.2)/0.6 (0.1) | 1.5 (0.5, 2.5) | 0.0042 |
| ERT-Lat | 6/788 | 1912.3 (559.9)/1994.6 (680.2) | 0.5 (−0.5, 1.5) | 0.35 |
| GAF | 9/2153 | 80.7 (4.6)/80.9 (10.0) | 0.2 (−0.7, 1.0) | 0.69 |

The cognitive assessment values are unadjusted means and SD. For analyses the cognitive test scores were inverse normal transformed and adjusted for sex, age and age². Cognitive test scores were then shifted and scaled, resulting in controls having a mean of 0 and a standard deviation of 1. The cognitive test scores were arranged such that higher scores indicate greater impairment, in *MAP1B* LoF carriers. The effects (β in SD) and P-values were calculated by comparing *MAP1B* LoF carriers (FAM1-B1,-C1,-C2,-D1,-D2,-D3, FAM2-H1,-J2, FAM3-L1) with controls using a generalised least-squares regression with a variance–covariance matrix based on the kinship coefficient of each pair of individuals. See Methods for a description of the cognitive tests
LMI&II: logical memory subtest from the Wechsler Memory Scale III (WMS-III), LF: letter fluency from the controlled oral word association test (COWAT), CF: category fluency from the category naming test, Stroop: name colour minus colour pad task, TMT: trail-making test (refers to part B minus part A), Pers. err.: perseverative error from the Wisconsin card-sorting test (WCST), SWM: spatial working memory, RVIP: rapid visual information processing, TMT-A: trail-making test part A, Str-bl: stroop black letter, ERT-Cor: emotion recognition task-percentage correct, ERT-Lat: emotion recognition task-mean response latency, SD: standard deviation, GAF: global assessment of functioning scale

**Truncated MAP1B protein is stable**. To determine whether the three *MAP1B* variants can produce a truncated MAP1B protein, we transiently overexpressed plasmids, incorporating each of the LoF mutations or wild-type *MAP1B*, in HeLa cells. We used western blot analysis, with antibodies recognising either the N-terminal or a V5 tag ligated to the C-terminal end of *MAP1B*, and detected bands in the expected sizes demonstrating that stable truncated proteins are generated from all three LoF variants (Fig. 4 and Supplementary Figure 7).

## Discussion

We used a family-based approach to find a LoF variant in *MAP1B* that associates with ID, GM and WM deficits and lower FA. The family-based approach scores coding-sequence variants, propagating in a pedigree, based on the predicted functional effect of the variant and disease co-segregation likelihood ratio. We estimated the genome-wide P-value using simulated segregation of coding variants, from the founders of a large sample of WGS Icelanders. Two additional stop-gain variants in *MAP1B* were found, in two independent families, which confirmed the association with cognitive impairments. A PGS-IQ is not significantly different in *MAP1B* carriers compared with controls, suggesting that the LoF variants are responsible for the IQ impairment.

MAP1B (reviewed in Ref. [31]) is a member of the microtubule binding family of proteins important for axonal growth and synapse maturation during brain development. *MAP1B* is primarily expressed in the neuronal soma, dendrites and axons where, during development, it is enriched in the axonal growth cone. In the adult brain *MAP1B* expression remains high in areas that retain plasticity, including olfactory bulb, hippocampus and cerebellum. MAP1B undergoes post-translational modification whereby it is cleaved into a heavy chain (HC) and a light chain (LC1) (Supplementary Figure 1).

Each MAP1B chain has microtubule and actin-binding domains that promote tubulin polymerisation and cross linking, and while we only established that our LoF variants could produce stable truncated protein, in-vitro studies have shown that similarly truncated MAP1B HC can promote microtubule assembly[32].

When phosphorylated, MAP1B maintains microtubules in a dynamic state required for growth cone-mediated axonal elongation. However, decreased MAP1B phosphorylation increases microtubule stability, leading to axon branching[33]. The majority of phosphorylation sites, regulating axonal growth and branching, are located downstream of the E712KfsTer10 *MAP1B* variant[34]. Loss of these sites, through MAP1B truncation, is therefore likely to result in impaired elongation and increased branching as

**Table 3 Brain white matter differences in *MAP1B* LoF carriers compared with controls**

| White matter | MAP1B-FS | Controls-FS | β (95% CI) | P |
|---|---|---|---|---|
| Corpus callosum—anterior—volume | 431.3 (155.9) | 900.2 (153.7) | −2.6 (−3.4, −1.9) | $1.9 \times 10^{-11}$ |
| Corpus callosum—posterior—volume | 472.9 (141.4) | 988.6 (156.7) | −2.5 (−3.3, −1.8) | $1.0 \times 10^{-10}$ |
| Corpus callosum—total—volume | 1689.3 (426.4) | 3210.3 (486.3) | −2.4 (−3.2, −1.7) | $5.5 \times 10^{-10}$ |
| LH—Cerebral white matter—volume | 190965.3 (23104.5) | 250084.3 (30115.5) | −2.2 (−2.9, −1.4) | $4.0 \times 10^{-08}$ |
| Total cerebral white matter—volume | 382905.6 (46892.5) | 502178.8 (60387.8) | −2.1 (−2.9, −1.4) | $5.1 \times 10^{-08}$ |
| RH—cerebral white matter—volume | 191940.3 (23831.4) | 252094.6 (30354.8) | −2.1 (−2.9, −1.3) | $6.8 \times 10^{-08}$ |
| Corpus callosum—central—volume | 268.2 (56.6) | 438 (87.7) | −1.9 (−2.7, −1.1) | $1.1 \times 10^{-06}$ |
| Corpus callosum—mid-posterior—volume | 245.2 (60.4) | 424.7 (92.3) | −1.8 (−2.6, −1.0) | $4.7 \times 10^{-06}$ |
| Corpus callosum—mid-anterior—volume | 271.8 (80.5) | 458.8 (103.6) | −1.8 (−2.5, −1.0) | $4.8 \times 10^{-06}$ |
| LH—white matter surface area | 75187.5 (8576.7) | 85866.9 (8321.5) | −1.5 (−2.3, −0.8) | $7.9 \times 10^{-05}$ |
| RH—white matter surface area | 75488.3 (8865.8) | 86285.6 (8411.9) | −1.5 (−2.3, −0.8) | $8.3 \times 10^{-05}$ |

The MAP1B-FS and Control-FS are unadjusted mean and SD MRI white matter volume ($mm^3$) and grey matter volume ($mm^3$), surface area ($mm^2$) and thickness (mm) values derived from FS. Individuals' FS values were inverse normal transformed and adjusted for sex, age, $age^2$, scanner model and ICV where appropriate. The effect (β in SD) and P-value were calculated by comparing *MAP1B* LoF carriers (n = 10; FAM1-B1,-C1,-C2,-D1,-D2,-D3, FAM2-H1,-J1,-J2, FAM3-L1) with controls (n = 949) using a generalised least-squares regression with a variance–covariance matrix based on the kinship coefficient of each pair of individuals. See Supplementary Data 1 for the full list of 274 traits. Bonferroni significance level was set at $0.05/274 = 1.8 \times 10^{-4}$
LH: left hemisphere, RH: right hemisphere, SD: standard deviation, FS: FreeSurfer

neurites from mouse *MAP1B* null models demonstrate[35] and is recapitulated in studies where a MAP1B kinase, glycogen synthase kinase 3β (GSK3β), a key regulator of axon growth and branching, is inhibited[36]. This could explain the observed lower FA in carriers as increased axonal branching or disordered arrangement may appear as a decrease in WM tract directionality.

The three *MAP1B* LoF mutations all result in absence of the LC1. LC1 has functions independent of the HC and separate from its role during neuronal development[31]. For instance, LC1 regulates the addition and removal of AMPA-type glutamate receptors at excitatory synapses, which forms one of the mechanisms of synaptic plasticity[37]. By forming immobile complexes, the LC1 traps AMPA receptors and limits intracellular vesicular trafficking within the dendritic shaft. Hence, a LoF mutation affecting the LC1 will prevent MAP1B from regulating and restricting the access of AMPA receptors to dendritic spines and the postsynaptic membrane. The *MAP1B* mutations, E712KfsTer10, E1032Ter and R1664Ter are therefore likely to affect the control of downregulation of synaptic transmission by increasing the availability of circulating neurotransmitter receptors[37]. Synaptic plasticity is accepted as the cellular basis of learning and memory, and the loss of this regulatory action could, in part, explain the carriers' cognitive impairments.

Expression and function of *MAP1B* is regulated by products of genes that when mutated, or lost, also result in neurodevelopmental disorders, including ID syndromes. A trinucleotide expansion polymorphism in fragile X mental retardation 1 (*FMR1*), results in the Fragile X syndrome (FXS), a common cause of ID[6] and autism[38]. *FMR1* codes for the fragile X mental retardation protein (FMRP), an RNA-binding protein and essential translation regulator of multiple genes during brain development, including *MAP1B*[38].

A recent study using FMR1-KO mice showed larger WM volume overall, mostly due to the larger CC[39]. This is supported by a human MRI study of adult males with FXS[40]. The loss of FMRP in fragile X neurons and consequent reduction in its translational suppression function leads to elevated MAP1B expression[38]. This may be a contributing factor to the increased WM, which is consistent with the phenotypes observed in the FXS cases and the mouse FMR1-KO models.

Chromosome 2p15p16.1 deletion syndrome, which harbours the zinc finger transcription factor *BCL11A-L*, is a rare genetic disorder characterised by physical and neurodevelopmental impairments, including ID, and structural brain abnormalities, including CC hypoplasia[41]. Knockdown of *BCL11A-L*, downregulates expression of *MAP1B* and the axon-guidance receptor

DCC (deleted in colorectal cancer), and induces axon branching, multi-axon formation, and dendrite outgrowth with suggestions that BCL11A-L plays an important role in neural circuit formation during development[42].

BCL11A-L and FMRP target numerous genes also important for neuronal development, and their loss affects *MAP1B* expression in opposite direction. However, it is clear that under- or over-connectivity can result in cognitive deficits and that *MAP1B* is a critical intermediary, demonstrated by the loss of BCL11A-L and FMRP.

Further research is required to unravel the effects of the truncated MAP1B-HC and loss of one copy of LC1 on brain development and synaptic plasticity. We have shown that LoF mutations produce stable truncated MAP1B and that LoF carriers develop both a neurological deficit and a cognitive impairment phenotype providing a comprehensive description of a syndrome involving *MAP1B*. LoF variants in *MAP1B* are found at a frequency of ~1 in 10,000, in publicly available databases, which places it within the range of the more common, single gene causes of syndromic ID.

## Methods

**Study population and cognitive assessment**. Samples, neuropsychological assessment and questionnaire data were collected as part of studies approved by the National Bioethics Committee in Iceland. Participants signed informed consent before buccal swabs or blood samples were drawn. Personal identifiers were encrypted by a third-party system overseen by the Icelandic Data Protection Authority[43].

Intellectual disability and autism diagnoses were assigned at the State Diagnostic and Counselling Centre (SDCC) using ICD10 criteria. Refer to Supplementary Table 1 for the Wechsler scale of intelligence available, from the SDCC, for each participant.

Phenotypic assessment, at the Study Recruitment Centre, as applied previously[17], uses the Mini-International Neuropsychiatric Interview (M.I.N.I.)[19] to assess psychiatric disorders and the Global Assessment of Functioning (GAF) to assess social and occupational functioning. The Icelandic version of the Wechsler Abbreviated Scale of Intelligence (WASI[IS])[44,45], which includes four subtests (Vocabulary, Similarities, Block Design and Matrix Reasoning), was administered to the *MAP1B* families and 1768 of the controls while another 458 controls were tested with two subtests, Vocabulary and Matrix Reasoning, from the Wechsler Adult Intelligence Scale (WAIS-III)[46]. As Icelandic norms are unavailable for WAIS-III, these scores were combined with WASI[IS] scores using general cognitive ability (an extracted g factor) as an indicator of the comparability of the two control groups. We also applied the logical memory subtest (abbreviated LMI&II) from the Wechsler Memory Scale III (WMS-III)[47], the controlled oral word association test (COWAT; here called LF)[48], the category naming test (here called category fluency)[49], the Stroop test (name colour minus colour pad task abbreviated Stroop and Stroop black letter)[50], the trail-making test (TMT refers to part B minus part A) and TMT-A)[51], the Wisconsin card-sorting test (WCST; perseverative error)[52], as well as the spatial working memory[53], rapid visual information processing[54] and emotion recognition task (ERT; percentage correct and mean response latency[55,56]

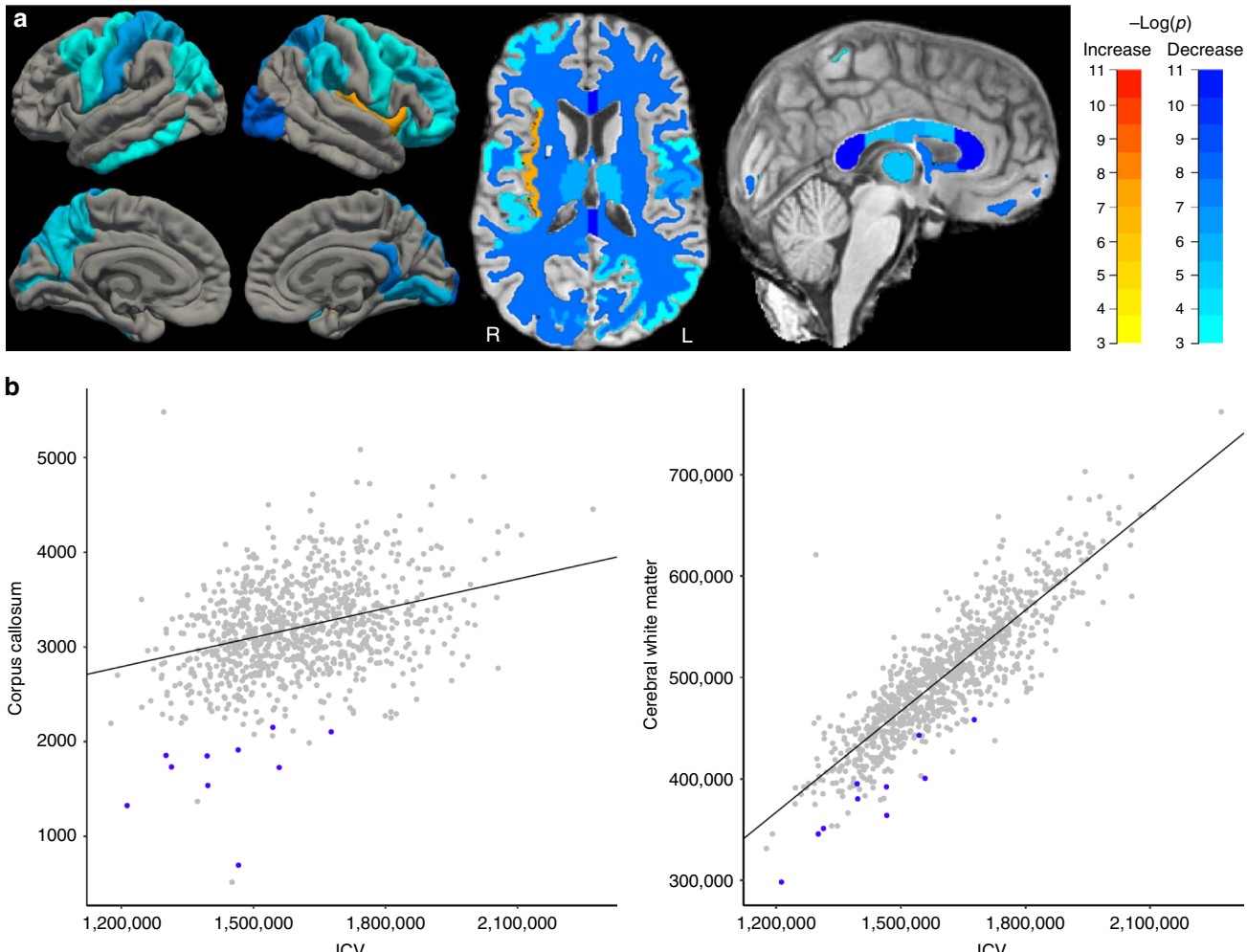

**Fig. 2** CC and WM volume differences in *MAP1B* LoF carriers compared with controls. **a** The colour scales refer to regions that are smaller (blue) or larger (red/yellow) in *MAP1B* LoF carriers ($n = 10$) compared with controls ($n = 949$). See Tables 3 and 4 for details. The four brain aspects in the left of the figure are from top left and clockwise: left hemisphere (LH) lateral surface, right hemisphere (RH) lateral surface, RH medial surface and LH medial surface. The middle image is an axial slice with the front of the brain at the top of the figure and demonstrates the ubiquitous nature of the WM and GM effects; R is right and L is left. The image on the right is a glass brain highlighting the significant reduction in WM in the corpus callosum. **b** Unadjusted corpus callosum and cerebral WM volumes plotted against intracranial volume (ICV) from FreeSurfer (in mm$^3$) for *MAP1B* LoF carriers (blue) compared with controls (grey). Relative to the linear regression (diagonal line) the *MAP1B* LoF carriers have smaller CC and cerebral WM volumes than would be predicted from their ICV

the last three are subtests of the Cambridge Neuropsychological Test Automated Battery (http://www.cambridgecognition.com/).

As the ERT was not included in the previously published assessment[17], it will be briefly described here. The CANTAB ERT is a modified version of the Bristol University Emotional Recognition Task[55,57]. A computer morphed face (aggregated from 12 males) is presented on screen, for 200 ms, expressing one of the six basic emotions (anger, disgust, fear, happiness, sadness or surprise)[56] at a level of intensity from 1 (least) to 15 (most). This is followed by a six-button alternative forced-choice, whereby the participant is required to choose which emotion was displayed on the previous image. Each emotion intensity combination is presented either once, for a total of 90 images or in each of two sets of 90 images for a total of 180 responses (6 emotions × 15 intensities × 2 rounds). Correct and incorrect responses and response time latency was recorded for each emotion intensity combination. ERT response time latency was averaged across the entire set of responses for an overall mean response latency and ERT correct responses were summed across all of the emotions and presented as a fraction of total number of correct responses, per individual.

In total 13 *MAP1B* LoF carriers had some form of Wechsler scale of intelligence assessment. Where available, for the *MAP1B* LoF carriers, the WASI$^{IS}$ was used for calculations, followed by WISC-IV and WPPSI-R (See Supplementary Table 1). Where PIQ was compared with VIQ using a one-sided, paired *t*-test, age and gender scaled IQ (mean of 100 and SD of 15) was used. Other IQ comparisons used IQ data that was inverse normally transformed. Values from the other cognitive tasks were inverse normally transformed, followed by adjustment for gender, age at

assessment and age[2], based on controls. Tests were standardised so that controls had a mean of zero and standard deviation of one and cognitive test scores arranged so that higher scores indicated greater impairment. To take the information on relatedness of the individuals into account, *MAP1B* LoF carriers (*n* varies per task, see Tables 1 and 2) were compared with controls (*n* > 2,200, see Tables 1 and 2) using generalised least-squares regression with a variance–covariance matrix based on the kinship coefficient of each pair of individuals (see Quantitative association replication and Reference[58]). To account for multiple comparisons, the Bonferroni correction was used, giving a significance threshold of $P = 0.05/20 = 2.5 \times 10^{-3}$ (seven WASI$^{IS}$ tests and 13 cognitive tests). We used RStudio (version 1.0.44; https://www.rstudio.com/) integrating R (version 3.3.2; https://www.r-project.org/) for data handling and some analyses, psych package (version 1.16.12) for some summary statistics and ggplot2 (version 2.2.1) to generate some of the figures.

**Sequencing.** Next-generation WGS, alignment and variant calling was carried out as described in ref. [59]. In brief, DNA samples, isolated from blood or buccal swabs, were prepared using TruSeq DNA, TruSeq Nano or TruSeq PCR-Free kits as per Illuminas instructions. DNA was fragmented, end-repaired, size-selected and purified, followed by PCR enrichment and sequencing libraries were assessed for quality and concentration. Following hybridisation of sequencing libraries to flowcells using the Illumina cBot, Illumina GAIIx, HiSeq 2,000/2,500 or HiSeq X instruments were used to perform sequencing-by-synthesis on paired-end libraries

**Table 4 Brain grey matter differences in *MAP1B* LoF carriers compared with controls**

| Grey matter | MAP1B-FS | Controls-FS | $\beta$ (95% CI) | $P$ |
|---|---|---|---|---|
| RH—lateral occipital—area | 3768.6 (336.8) | 4813.9 (618.9) | −2.1 (−2.9, −1.3) | $1.1 \times 10^{-07}$ |
| RH—isthmus cingulate—volume | 1690.8 (287.0) | 2406.6 (426.9) | −2.0 (−2.8, −1.3) | $2.2 \times 10^{-07}$ |
| Right thalamus proper—volume | 6056.4 (720.9) | 7099.5 (756.2) | −1.9 (−2.7, −1.2) | $7.9 \times 10^{-07}$ |
| Total thalamus—volume | 12720.9 (1506.2) | 15006.4 (1608.0) | −1.9 (−2.7, −1.2) | $1.1 \times 10^{-06}$ |
| RH—superior parietal—volume | 10269.2 (1554.5) | 12473.5 (1572.3) | −1.9 (−2.7, −1.1) | $1.1 \times 10^{-06}$ |
| LH—postcentral—area | 3433.9 (387.5) | 4193.3 (502.1) | −1.9 (−2.7, −1.1) | $1.3 \times 10^{-06}$ |
| RH—insula—thickness | 3.3 (0.1) | 3.0 (0.2) | 1.9 (1.1, 2.6) | $1.9 \times 10^{-06}$ |
| RH—isthmus cingulate—area | 697.0 (83.3) | 962.5 (165.6) | −1.9 (−2.6, −1.1) | $2.0 \times 10^{-06}$ |
| LH—superior parietal—volume | 10221.1 (1301.4) | 12567.8 (1661.6) | −1.8 (−2.5, −1.0) | $4.9 \times 10^{-06}$ |
| RH—lingual—volume | 5014.4 (689.2) | 6379.8 (911.4) | −1.8 (−2.6, −1.0) | $5.3 \times 10^{-06}$ |
| LH—postcentral—volume | 7424.0 (727.9) | 9164.1 (1327.5) | −1.7 (−2.5, −1.0) | $9.3 \times 10^{-06}$ |
| Left thalamus proper—volume | 6664.5 (838.1) | 7906.9 (930.3) | −1.7 (−2.5, −1.0) | $1.1 \times 10^{-05}$ |
| RH—lateral orbito frontal—area | 2247.6 (287.6) | 2678.3 (312.4) | −1.7 (−2.4, −0.9) | $1.9 \times 10^{-05}$ |
| RH—rostral middle frontal—area | 5043.1 (825.1) | 5987.8 (778.0) | −1.7 (−2.4, −0.9) | $2.2 \times 10^{-05}$ |
| RH—lateral occipital—volume | 9177.2 (1004.8) | 11253.2 (1607.7) | −1.6 (−2.4, −0.9) | $2.4 \times 10^{-05}$ |
| LH—pericalcarine—area | 1025.0 (208.3) | 1431.4 (240.6) | −1.6 (−2.4, −0.9) | $2.5 \times 10^{-05}$ |
| RH—supramarginal—volume | 8493.2 (1189.4) | 9937 (1474.5) | −1.6 (−2.4, −0.9) | $3.5 \times 10^{-05}$ |
| LH—precuneus—volume | 7665.6 (1201.7) | 9202.4 (1259.4) | −1.6 (−2.4, −0.8) | $4.0 \times 10^{-05}$ |
| LH—superior parietal—area | 4524.4 (632.9) | 5480.6 (652.4) | −1.6 (−2.3, −0.8) | $5.3 \times 10^{-05}$ |
| LH—caudal middle frontal—volume | 5046.0 (1235) | 6265.4 (1168.2) | −1.6 (−2.3, −0.8) | $5.6 \times 10^{-05}$ |
| LH—caudal middle frontal—area | 1872.9 (395.3) | 2357.7 (386.6) | −1.6 (−2.3, −0.8) | $6.9 \times 10^{-05}$ |
| RH—supramarginal—area | 3140.5 (543.8) | 3717.1 (510.1) | −1.6 (−2.3, −0.8) | $8.6 \times 10^{-05}$ |
| LH—inferior temporal—area | 2853.9 (423.8) | 3477.8 (467.4) | −1.5 (−2.3, −0.8) | $8.8 \times 10^{-05}$ |
| LH—inferior parietal—area | 3881.0 (729.8) | 4648.8 (634.0) | −1.5 (−2.3, −0.7) | $1.3 \times 10^{-04}$ |
| RH—parsorbitalis—area | 675.0 (98.6) | 778.9 (107.8) | −1.5 (−2.3, −0.7) | $1.3 \times 10^{-04}$ |
| LH—inferior parietal—volume | 10169.3 (1777.0) | 12202.0 (1838.4) | −1.5 (−2.3, −0.7) | $1.4 \times 10^{-04}$ |
| RH—caudal middle frontal—area | 1700.7 (328.7) | 2199.7 (383.0) | −1.5 (−2.3, −0.7) | $1.5 \times 10^{-04}$ |
| RH—precentral—area | 4096.8 (569.3) | 4820.5 (574.8) | −1.5 (−2.3, −0.7) | $1.6 \times 10^{-04}$ |
| RH—lingual—area | 2548.2 (274.5) | 3175.1 (396.4) | −1.5 (−2.2, −0.7) | $1.7 \times 10^{-04}$ |

The MAP1B-FS and Control-FS are unadjusted mean and standard deviation (SD) MRI white matter volume ($mm^3$) and grey matter volume ($mm^3$), surface area ($mm^2$) and thickness (mm) values derived from FreeSurfer (FS). Individuals' FS values were inverse normal transformed and adjusted for sex, age, $age^2$, scanner model and ICV where appropriate. The effect ($\beta$ in SD) and $P$-value were calculated by comparing *MAP1B* LoF carriers ($n = 10$; FAM1-B1,-C1,-C2,-D1,-D2,-D3, FAM2-H1,-J1,-J2, FAM3-L1) with controls ($n = 949$) using a generalised least-squares regression with a variance-covariance matrix based on the kinship coefficient of each pair of individuals. See Supplementary Data 1 for the full list of 274 traits. Bonferroni significance level was set at $0.05/274 = 1.8 \times 10^{-4}$
LH: left hemisphere, RH: right hemisphere, SD: standard deviation, FS: FreeSurfer

followed by imaging and base-calling in real-time. All of the *MAP1B* carriers were sequenced on the HiSeq X instrument. Reads were aligned to NCBI Build 38 of the human reference sequence and then merged into a single BAM file, followed by multi-sample variant calling performed with GenomeAnalysisTK (GATK) version 2.3.9.

Applied Biosystems BigDye Terminator v3.1 Cycle Sequencing Kit was used to confirm *MAP1B* LoF carrier status. PCR and cycle sequencing reactions were performed on MJ Research PTC-225 thermal cyclers. Applied Biosystems 3730xl DNA analyzers were used for signal detection.

All positions reported are NCBI human genome build 38 unless otherwise stated.

**Family-based association analysis**. We applied a recently described family-based method[12] to test rare coding-sequence variants for disease segregation within the pedigree. In order to try and account for the extreme clustering of cases in the pedigree we directed our search towards high-penetrance coding variants carried by less than 30 whole-genome sequenced individuals. To test for association, we create a scoring function based on the effect of the coding variant, as well as its co-segregation with the disease, both inside and outside of the pedigree. Genome-wide simulations are then used to provide significance estimates.

We used logistic regression, applying sex, year of birth, county of birth and lifespan as covariates in the whole dataset, to estimate the probability of individual $i$ being affected: let $A_i$ be the affection status of individual $i$ and $x_i$ be his covariates, then we estimate

$$f_i = \text{logit}(p(A_i|x_i)),$$

or equivalently

$$p(A_i|x_i) = \frac{e^{f_i}}{1 + e^{f_i}}.$$

Our analysis tests particular variants, in a known set of carriers, distinct from parametric linkage analysis which searches for linkage with unknown linkage variants. However, comparable to parametric linkage[60], the variant being tested has an assumed effect.

We assume that the genetic effect parameter $\beta$, for the genotype $g_i$ of individual $i$, affects penetrance as follows:

$$\text{logit}(p(A_i|g_i, x_i)) = f_i + \beta g_i$$

or equivalently

$$p(A_i|g_i, x_i) = \frac{e^{f_i + \beta g_i}}{1 + e^{f_i + \beta g_i}}.$$

We assumed $g_i$ to be either 0 or 1 given the dominant mode of inheritance pattern of the cases. We used a peeling algorithm[61], to calculate the likelihood of the observed cases and controls, and so to estimate the genetic effect of the supposed disease causing variant behind the familial clustering. We then obtained a maximum likelihood estimate of the genetic effect $\beta$ by maximising the likelihood with a Broyden–Fletcher–Goldfarb–Shannon (BFGS) maximizer.

We then applied the optimal test statistic, to test whether a genetic variant is causal or has no effect on disease, as stated by the likelihood ratio principle:

$$\begin{aligned}
\text{LLR} &= \log \frac{p(A|x, \beta = \hat{\beta})}{p(A|x, \beta = 0)} = \sum_i \log\left(p\left(A_i | g_i, x_i, \beta = \hat{\beta}\right)\right) \\
&\quad - \log(p(A_i|g_i, x_i, \beta = 0)) \\
&= \sum_{i:A_i=1} \log\left(\frac{e^{f_i + \beta g_i}}{1 + e^{f_i + \beta g_i}}\right) - \log\left(\frac{e^{f_i}}{1 + e^{f_i}}\right) \\
&\quad + \sum_{i:A_i=0} \log\left(\frac{1}{1 + e^{f_i + \beta g_i}}\right) - \log\left(\frac{1}{1 + e^{f_i}}\right) \\
&= \sum_{i:A_i=1} g_i \left(\log\left(\frac{e^{f_i + \hat{\beta}}}{1 + e^{f_i + \hat{\beta}}}\right) - \log\left(\frac{e^{f_i}}{1 + e^{f_i}}\right)\right) \\
&\quad + \sum_{i:A_i=0} g_i \left(\log\left(\frac{1}{1 + e^{f_i + \hat{\beta}}}\right) - \log\left(\frac{1}{1 + e^{f_i}}\right)\right).
\end{aligned}$$

While affected and unaffected carriers have a positive and negative impact, respectively, the log likelihood ratio terms are 0 for noncarriers. In order to avoid blanket removal of variants not unique to the pedigree and to upweigh variants

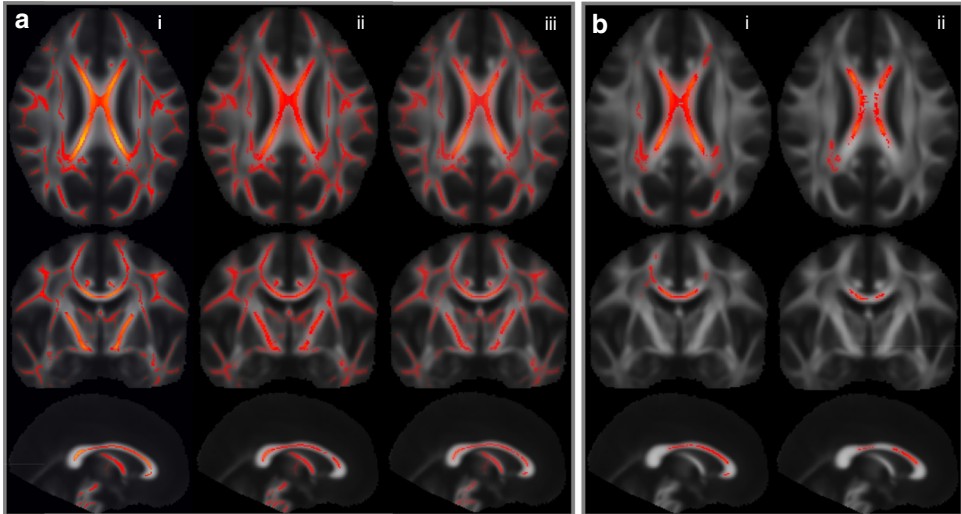

**Fig. 3** Diffusion (TBSS) data mapping from *MAP1B* LoF carriers compared with controls. Rows one, two, and three show, respectively, axial, coronal, and sagittal aspects of brain (MNI coordinates: 1, −13, 19) where heat map (red/yellow) intensity represents a significant change in fractional anisotropy (FA). **a** We compared *MAP1B* LoF carriers ($n = 9$) with three control groups (column i: normal range controls ($n = 181$), column ii: controls with small CC volume ($\beta < -1.5$ SD, $n = 15$), and column iii: controls with small WM volume ($\beta < -1.5$ SD, $n = 10$)) and found the carriers to have significantly lower brain-wide FA compared with all three control groups. **b** To determine whether a smaller CC volume generally leads to a decrease in FA, we compared the, column (i) 15 controls with smaller CC with 166 normal range controls and, column (ii) the ten controls with smaller WM with 160 normal range controls and found a reduction in FA only in the densest part of the CC. Family wise error (FWE) corrected significance threshold of 0.05

present in affecteds outside the pedigree, we counted carriers outside the family in the score.

We define the score for marker $m$ as

$$\text{LLR}(m) - \log(P(\text{coding effect of } m)) \cdot$$

We are only considering coding variants and $-\log(P(\text{coding effect of } m))$ will be 2.96 higher for loss-of-function variants than moderate impact variants based on the frequencies of such coding mutations[62].

To estimate the significance of a variant we used genome-wide simulations, sampling founders from a set of 30 K whole-genome sequenced Icelanders outside of the pedigree and counting the number of times the highest score was equal to or higher than the observed score. From 100,000 simulations a distribution of the highest score for all markers in the genome could be generated and a genome-wide $P$-value estimated from the fraction of simulations with scores greater than or equal to the highest observed score. We took $P < 0.05$ to be significant.

**Case control association replication.** Logistic regression was used to test for association between variants and disease, treating disease status as the response and genotype counts (alleles from either LoF variant) as covariates (explanatory variables). We also included sex, county of birth, current age or age at death (first and second order terms included), blood sample availability for the individual and an indicator function for the overlap of the lifetime of the individual with the timespan of phenotype collection as nuisance variables in the model.

Given genotype counts for $n$ individuals, $g_1, g_e, \ldots, g_n \epsilon \{0,1,2\}$, their phenotypes $y_1, y_2, \ldots, y_n \epsilon \{0,1\}$ and a list of vectors of nuisance parameters $x_1, x_2, \ldots, x_n$, the logistic regression model states that

$$L_i(\alpha, \beta, \gamma) = P(y_i = 1 | g_i, x_i)$$

$$\text{logit}(P(y_i = 1 | g_i, x_i)) = \alpha + \beta g_i + \gamma^T x_i, \text{ for all } i \in \{1, 2, \ldots, n\},$$

where $\alpha$, $\beta$ and $\gamma$ are the regression coefficients and $L_i$ is the contribution of the $i$ th individual to the likelihood function; $L(\alpha, \beta, \gamma) = \prod_{i=1}^{n} L_i(\alpha, \beta, \gamma)$. It is then possible to test for association based on the asymptotic assumption that the likelihood ratio statistic follows a $\chi^2$ distribution with one degree of freedom:

$$2 \log \left( \frac{\max\limits_{\alpha, \beta, \gamma} L(\alpha, \beta, \gamma)}{\max\limits_{\alpha, \gamma} L(\alpha, \beta = 0, \gamma)} \right) \sim \chi_1^2.$$

For the replication only whole-genome sequenced cases and controls were used; of those FAM2-J1 and J3 are case carriers and FAM2-H1,-J2 and FAM3-L1 are control carriers.

**Quantitative association replication.** A generalised form of linear regression was used to test IQ for association with the variants.

The regression model assumes a normal distribution of the quantitative measurements ($y$) with a mean that depends linearly on the expected allele ($g$) of the variant being tested and a variance–covariance matrix proportional to the kinship matrix:

$$y \sim N(\alpha + \beta g, 2\sigma^2 \phi),$$

Where

$$\phi_{ij} = \begin{cases} \frac{1}{2}, i = j \\ 2k_{ij}, i \neq j \end{cases}$$

is the kinship matrix as estimated from the Icelandic genealogical database.

**Genetic analysis.** Microarray genotyping and long-range phasing was described previously[59]. Briefly, chip-typed samples are assayed with an Illumina bead chip at deCODE genetics. Chip SNPs are excluded based on a number of quality criteria, and samples with a call rate below 97% are not used.

Long-range phasing to determine parent of origin was achieved as outlined in ref. [63]. In brief, assigning the parent of origin was performed by identifying the closest relatives who shared a haplotype with the proband. Parental origin was then assigned to the two haplotypes of a proband on the basis of a computed score.

**Computing IQ PGS and effect on IQ.** Summary statistics from a recent GWAS for IQ[23] and educational attainment[24] provided the per-locus allele-specific weightings to calculate a PGS-IQ and PGS-EDU, respectively. The basic method used to process the genotype data for Icelanders, including imputations based on full-genome sequencing results, was described above and in Reference[59]. The markers and methods used to compute the PGS-IQ and PGS-EDU have previously been described for computing an educational attainment PGS in Kong et al[22]. A framework set of ~600,000 high-quality SNPs (not including any of the *MAP1B* LoF variants) covering the whole genome was used to compute PGS-IQ and PGS-EDU. We adjusted for linkage disequilibrium using LDpred[25]. The linkage disequilibrium between markers was estimated using the Icelandic samples. We used Welch two sample t-test to compare PGS-IQ and PGS-EDU in *MAP1B* carriers ($n = 13$) to controls ($n = 2,164$) in R (stats version 3.3.2).

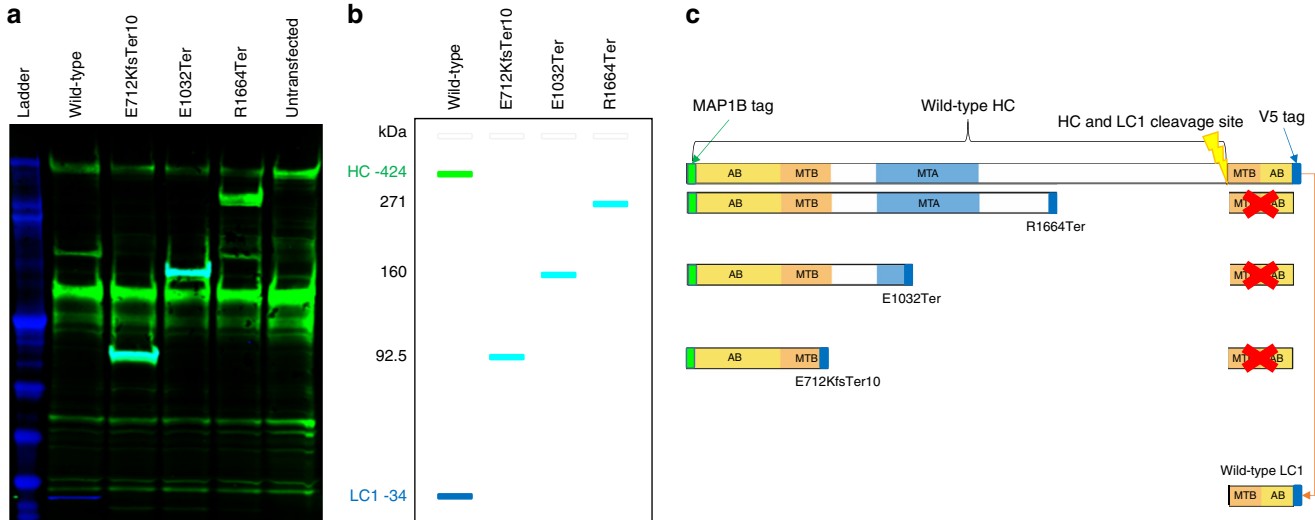

**Fig. 4** Western blot of antibody binding to wild-type and truncated MAP1B proteins. **a** An image, from the Odyssey system, where the colour coded infrared fluorescent signals from the 700 nm channel (V5 antibody signal—blue) and the 800 nm channel (MAP1B antibody signal—green) are taken from the same blot; cyan indicates where both antibody signals overlap. See Supplementary Figure 7 for the individual V5 and MAP1B antibody signal greyscale pictures. **b** A schematic diagram of what each band on the western blot represents in the context of protein size (heavy chain (HC), light chain 1 (LC1) and truncated proteins). **c** A schematic diagram of truncated MAP1B and an indication of whether the anti-V5 or anti-MAP1B antibodies bind the protein product. The wild-type HC is only recognised by the anti-MAP1B antibody (green), and the wild-type LC1 is only recognised by the anti-V5 antibody (blue), while both anti-MAP1B and anti-V5 antibodies bind to all three truncated protein products as they contain both the MAP1B (amino acids 6–31) and V5 tags (cyan bands in **b**). The red X indicates that the LC1 is not produced as a consequence of the MAP1B truncation. AB actin-binding domain, MTA microtubule assembly helping site, putative, MTB microtubule-binding domain

**Magnetic resonance imaging acquisition**. Magnetic resonance imaging (MRI), previously described[17,64] but summarised here, was performed on two 1.5 T whole body scanners at Röntgen Domus Medica, a Philips Achieva and a Siemens Magnetom Aera. Scans were done with a sagittal 3D fast T1-weighted gradient echo sequence (Philips Achieva: repetition time (TR) = 8.6 ms, echo time (TE) = 4.0 ms, flip angle (FA) = 8°, 170 slices, slice thickness = 1.2 mm, acquisition matrix = 192 × 192, field of view (FOV) = 240 × 240 mm; Siemens Aera: repetition time (TR) = 2400 ms, echo time (TE) = 3.54 ms, flip angle (FA) = 8°, 160 slices, slice thickness = 1.2 mm, acquisition matrix = 192 × 192, field of view (FOV) = 240 × 240 mm). Ten *MAP1B* LoF carriers (on the Siemens Magnetom Aera) and 949 control subjects were scanned (257 with the Siemens Magnetom Aera and 692 with the Philips Achieva). Prior to FreeSurfer analyses, T1-weighted image data was converted from DICOM to NIfTI format.

Diffusion tensor image (DTI) data was acquired on the Siemens Magnetom Aera 1.5 T system. Diffusion-weighted echo planar imaging (EPI) was used. Three series of 12 non-co-linear gradient diffusion-weighted images at $b = 800$ S/mm$^2$ and one non-weighted ($b = 0$ S/mm$^2$) image was acquired, for a total of 39 images. The parameters for each image were as follows: TR = 8600 ms, TE = 60 ms, 60 slices, slice thickness = 2 mm, acquisition matrix = 120 × 120, FOV = 240 × 240 mm, resulting in data acquired with a 2 × 2 × 2 mm voxel resolution.

**Clinical MRI evaluation**. Along with the sagittal 3D fast T1-weighted gradient echo sequence, scans were also done with an axial proton density T2-weighted turbo spin echo sequence (Siemens Aera: TR = 3000 ms, TE-1 = 12 ms, TE-2 = 96 ms, FA = 150°, 48 slices, slice thickness = 3 mm, acquisition matrix = 256 × 232, FOV = 240 × 240 mm). Prior to clinical assessment, T1- and T2-weighted image data was converted from DICOM to NIfTI format. The neuroradiologist was unaware of *MAP1B* LoF carrier status and was provided with equal numbers of noncarriers, prior to evaluating symptoms of periventricular nodular heterotopia. T1- and T2-weighted images were viewed using MRIcron version 12.12.2012 (http://www.mricro.com).

**Surface-based morphometry**. Segmentation and parcellation of the T1-weighted image data was performed with FreeSurfer V5.3.0 software (http://freesurfer.net)[65]. FreeSurfer automatically identifies cerebrospinal fluid (CSF), white matter (WM)[66] and GM tissue in the brain and automatically segments and parcellates regions according to predefined atlases; cortical regions according to the Desikan/Killiany Atlas[67] and subcortical regions according to the Fischl/Salat Atlas[68]. FreeSurfer extracts region of interest (ROI) volumes (mm$^3$) for all defined subcortical regions and cortical thickness (mm), surface area (mm$^2$) and volume (mm$^3$) for all cortical

ROIs. This provided 104 Desikan/Killiany Atlas (times two for LH and RHs) cortical values for analysis. Combined values for certain volumes were also created. A total of 274 volumes, surface areas and thickness values were analysed (Supplementary Data 1).

Volume, surface area and cortical thickness values were inverse normally transformed, followed by adjustment for gender, age at scan, age$^2$, scanner type and where applicable ICV, based on data from controls only; intracranial volume (ICV) was used as an adjustment variable for volume (except ICV was not adjusted for itself) and surface area, but not for thickness. To take the information on relatedness of the individuals into account, *MAP1B* LoF carriers ($n = 10$) were compared with controls ($n = 949$) using generalised least-squares regression with a variance–covariance matrix based on the kinship coefficient of each pair of individuals (see Quantitative association replication and Ref. [58]). To account for multiple comparisons, the Bonferroni correction was used, giving a significance threshold of $P = 0.05/274 = 1.8 \times 10^{-4}$.

**Diffusion tensor imaging**. The diffusion-weighted data was preprocessed using the FMRIB Software Library (FSL)[69]. First, FSL's diffusion toolbox (FDT) was used for eddy current and motion correction, where the eddy currents are corrected with an affine registration to the b0 image and motion correction with a rigid body registration to b0. Second, non-brain areas were removed from the analysis using the BET tool[70] from FSL. Finally, FDT was used to fit the tensors at each voxel and create FA maps for each subject.

**Tract-based spatial statistical analysis**. Tract-based spatial statistics (TBSS) was used to perform voxel-wise statistical analysis of the FA data[71]. FA data were aligned into a common space using the nonlinear registration tool FNIRT[72,73], which uses a b-spline representation of the registration warp field[74], for all subjects. This was followed by creating a mean FA skeleton, which represents the centres of all tracts common to the group, from the thinned, mean FA image. Data, resulting from each subject's aligned FA data projected onto the mean FA skeleton, were then fed into voxel-wise cross-subject statistics.

Multiple regression models that included age and gender as covariates of no interest were used to investigate the effect of *MAP1B* on FA. A comparison between controls and *MAP1B* subjects (CONTROL > MAP1B and MAP1B < CONTROL) was performed. Additionally, *MAP1B* subjects were compared with CONTROLs with lower WM volume and CONTROLs with smaller CC. Statistical significance was assessed using FSL's randomise function[75] with the threshold-free cluster enhancement (TFCE). The significance assessment was based on 5000 random permutations and a corrected $P < 0.05$ was considered as significant.

**Generation of MAP1B plasmid variants**. We obtained an ORF Flexi clone from Promega (product ID FXC02218) that contained full-length *MAP1B* cDNA (GenBank: NM_005909). From that clone, full-length *MAP1B* cDNA was amplified by PCR primers F-5′ATGGCGACCGTGGTGGTG′3, R-5′CAGTTCAATCTTG CATGCAGGG′3 and cloned into pcDNA3.1/V5-His TOPO vector (Invitrogen K4800-01) following manufacturers protocol resulting in pcDNA3.1_MAP1B_WT. Transformed TOP10 chemically competent cells (Invitrogen C4040-10) were plated on LB plates containing 100 μg/ml ampicillin. Colonies were expanded in LB medium containing 100 μg/ml ampicillin. Plasmids were purified using Monarch nucleic acid purification kit (New England Biolabs T1010L) following the manufacturers protocol. Plasmid sequence was confirmed by Sanger sequencing.

In order to generate *MAP1B* cDNA variants E1032Ter and R1664Ter the pcDNA3.1_MAP1B_WT plasmid was used as a template. In short, a PCR reaction was performed using the following primers E1032Ter (F-5′ATGGCGACCGTGG TGGTG′3, R-5′CTCTCTGGCATCTTCAGCTTTG′3), and R1664Ter (F-5′ATG GCGACCGTGGTGGTG′3, R-5′ACTGAAGTCCATAGCAAGGGAT′3) resulting in a double stranded DNA fragment representing the entire pcDNA3.1 plasmid and the first 1031, and 1663 amino acids of *MAP1B* for the E1032Ter, R1664Ter variants, respectively. The STOP codon of the pcDNA3.1 plasmid was utilised instead of the one encoded by the *MAP1B* variants in order to attach a 3′ V5-tag to the protein variants in case *MAP1B* antibodies would not be able to recognise the variant protein product.

To generate the E712KfsTer10 variant, we first mutated the pcDNA3.1_ MAP1B_WT plasmid using the Q5 site-directed mutagenesis kit (New England BioLabs E0554S) and the following primers (F-5′AAGTTAAGAAGGAAGAGA AG′3, R-5′CTTCTTGACTTCCTTTGG′3) generating pcDNA3.1_MAP1B_ WT_2133delG. With this mutagenesis, the *MAP1B* protein will be correct from amino acid 1-711 followed by nine out of frame amino acids (KLRRKRRRK) plus a termination codon. We subjected this mutagenesis product to a PCR to get a template that contains the initiation codon of *MAP1B* in addition to the following 719 AA minus the stop codon using the following primers (F-5′ATGGCGACC GTGGTGGTG′3, R-5′CTTCCTTCTTCTCTTCCTTCTTAAC′3). This template was then cloned into pcDNA3.1 in frame with the V5 tag as described for the *MAP1B* E1032Ter and R1664Ter mutations, resulting in pcDNA3.1_MAP1B_ E712KfsTer10. The sequences of pcDNA3.1_MAP1B_E712KfsTer10, pcDNA3.1_ MAP1B_E1032Ter, and pcDNA3.1_MAP1B_R1664Ter were confirmed by Sanger sequencing.

**Expression of MAP1B in cultured cells**. One day prior to transfection, 320,000 HeLa cells (Public Health England 93021013) were seeded into each well of a 6-well plate in 3 mL of DMEM medium (11995-065, ThermoFisher) supplemented with 10% foetal calf serum (ThermoFisher 10500-064) and 50 units/mL penicillin and 50 μg/mL streptomycin (ThermoFisher 15070-063). On the day of transfection, media was replaced with the identical media as before without antibiotics. On the day of transfection, for each transfected well, 2.5 μg of pcDNA3.1_MAP1B_WT, pcDNA3.1_MAP1B_E712KfsTer10, pcDNA3.1_MAP1B_E1032Ter, or pcDNA3.1_MAP1B_R1664Ter plasmids were diluted in 125 μL Opti-Mem medium (ThermoFisher 31985- 047) and 5 μL of P3000 reagent (ThermoFisher L3000-008). Next, 5 μL Lipofectamine 3000 (ThermoFisher L3000-008) was mixed with 125 μL of Opti-Mem. Subsequently, the diluted plasmid solution was mixed with the Lipofectamine 3000 solution at a 1:1 ratio and incubated at room temperature for 12 min before the addition of 250 μL of the combined solution to each transfected well. Forty-eight hours post transfection, cell media was changed to full media and 24 h later cells were harvested for analysis.

**Western blot analysis**. Wells were washed 2× with PBS and cells corresponding to one well of a 6-well plate were lysed using 150 μl of RIPA buffer with 1:100 Halt protease and phosphatase inhibitor cocktail (Thermo Scientific 78442). Lysates were kept on ice for 10 min with agitation followed by sonication for 1 min (Branson 2510). Lysates were spun down at 4 °C for 15 min at 14,000× g. Total amount of protein lysates was estimated using BCA protein assay (Thermo Scientific 23227). Samples were prepared using Novex Bolt LDS sample buffer (Life technologies B0007) and Novex Bolt sample reducing agent (Life technologies B0009) and run on NuPage 3–8% Tris-Acetate gel (ThermoFisher EAO375BOX). Total protein amount per lane was 60 μg and HiMark Pre-stained HMW protein standard (ThermoFisher Cat: LC5699) was used to estimate protein size. Gel was run at a constant 150 V for 1 hour. Proteins were transferred to a PVDF membrane using wet blot transfer (mini trans blot BioRad) at a constant 70 V for 3 hours. Membranes were allowed to dry and then hydrated with MeOH and MQ water before blotting. Membranes were blocked for 1 hour at room temperature using 3% BSA in 1× TBS. Primary antibodies used were α-*MAP1B* (Santa Cruz sc-365668) 1:500 (recognises amino acids 6-31) and α-V5 (R&D MAB8926) 1:1000 incubated in blocking buffer with the addition of 0.1% Tween overnight at + 4 °C. Secondary antibodies used were α-Rabbit 680RD (Li-Cor 926-68073) and α-Mouse 800CW (Li-Cor 926-32212) both 1:20,000 in TBST + 0.01% SDS for 1 hour at room temperature. After washing the membrane, it was allowed to dry and then scanned using the Odyssey infrared imaging system (Li-Cor Biosciences). Anti-V5

and anti-MAP1B (amino acids 6-31) antibodies were visualised at 700 nm and 800 nm wavelengths, respectively.

**URL**. For American Psychiatric Association (APA), see https://psychiatry.org. For Exome Aggregation Consortium (ExAc), see http://exac.broadinstitute.org/gene/ENSG00000131711.

**Data Availability**. Data from publicly available databases (ExAc, gnomAD, 1000Genomes, NCBI-dbSNP/VarView, EVS-ESP, DECIPHER) and from the Simons Simplex Collection approved researcher access, was presented in the main text and supplementary material of this publication, but not used in any analyses. The data used in the analyses for this publication are not publicly available due to information, contained within them, that could compromise research participant privacy. The authors declare that the data supporting the findings of this study are available within the article, its Supplementary Information and upon request.

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

## Acknowledgements

We are grateful to the participants and we thank the psychologists, nurses and staff, in particular Berglind Eiriksdottir, at the Research Recruitment Center and technicians and staff at Röntgen Domus. We also thank the staff at deCODE genetics core facilities and all our colleagues for their important contribution to this work. L.J. received support from the Swedish Society of Medicine, the Swedish Brain Foundation and Swedish Society for Medical Research. The research leading to these results has received support from the Innovative Medicines Initiative Joint Undertaking under grant agreements' no. 115008 (NEWMEDS) and no. 115300 (EUAIMS) of which resources are composed of EFPIA in-kind contribution and financial contribution from the European Union's Seventh Framework Programme (EU-FP7/2007-2013), EU-FP7 funded grant no. 602450 (IMAGEMEND) and EU funded FP7-People-2011-IAPP grant agreement no. 286213 (PsychDPC).

## Author Contributions

G.B.W., O.G., G.S., V.K.E., G.A.J., S.S., D.F.G., M.O.U., E.S., H.S. and K.S. designed the study. G.B.W., O.G., G.A.J., U.U., A.k., L.J., M.S.N., A.L. and H.S. were involved in informatics and data management. G.B.W., V.K.E., G.A.J., E.S. and H.S. carried out cohort ascertainment, recruitment and phenotyping. A.d.J., A.Si. and A.s.J. performed the genotyping. A.B.A., A.L.L. and G.L.N. performed the laboratory experiments. G.B.W., O.G., M.I.M., M.O.U. and H.S. managed and analysed the MRI data. A.L. performed the MRI clinical assessment. G.B.W., O.G., G.S., S.S., A.F.G., P.S., M.F., A.I., D.F.G., M.O.U. and H.S. provided statistical methods and performed analyses. G.B.W., O.G., G.S., A.B.A., M.Z., D.F.G., M.O.U., H.S. and K.S. wrote the manuscript with contributions to the final version from all authors.

## Additional information

**Competing interests:** G.B.W., O.G., G.S., A.B.A., G.A.J., S.S., A.F.G., M.I.M., U.U., A.L.L., A.d.J., A.Si., A.s.J., A.Sk., M.S.N., P.S., M.F., A.I., G.L.N., M.Z., D.F.G., M.O.U., H. S. and K.S. are employees of deCODE genetics/Amgen. The remaining authors declare no competing interests.

