## [peer review file · Nature Communications]

Reviewer #1 (Remarks to the Author):

The authors present a WGS study identifying a frameshift variant in MAP1B that is present in 8 members of a single family 7 of whom are affected. Two other families were identified with stop-gain variants in MAP1B with carriers in both families showing patterns of intellectual impairment. Carriers of these variants showed lower white matter in MRI with these changes concentrated in the corpus callosum.

The paper is clearly written and presents an important finding that will be of interest to a diverse group of readers.

Specific comments.

1) Page 4 line 70 - Please add a sentence commenting on the 8th family member (A1?). There is some information in the supplement and I'm guess the AD is the reason for the lack of IQ data however it would be helpful to state this. For readers who only read the main text it will appear that the 8th family member is unaffected.

2) Can you also please add a comment on individual D4 who is not a carrier but has severe ID?

3) Given WGS data are available for these individuals it would be very helpful to know where they sit on a polygenic risk score distribution for IQ as compared to the controls. Having information about their burden of genetic risk due to common variants would help quantify the extent to which these deficits are due solely to the MAP1B variants vs their polygenic background risk. Multiple non-carriers in this family also appear to have deficits that are presumably due to the burden of common risk variants.

4) Methods - Many of the analyses state that country of birth was included as a covariate. It's unclear if this is to correct for ancestry or if all participants are of Icelandic ancestry and this is to correct for a cultural effect relating to migration. Please clarify. Also if this is a correction for ancestry please elaborate on how this is a sufficient correction.

5) Table 1 - Please provide the mean and se|sd on the raw scale for each test

6) Table 2 - Please provide the se|sd for each of the imaging measures to accompany these means.

Minor Text suggestions

Page 3 line 68 - change "truncation () associated" -> "truncation () was associated"

Page 6 line 116 - change "and less brain-wide" -> "and lower brain-wide"

Page 9 lines 193-198. Consider breaking this sentence at the point where you transition from talking about the human MRI studies to the mouse knockouts. Also please check/clarify the section of this sentence talking about the mouse studies. It is unclear what is meant by "significantly larger total brain volume differences in major WM structures throughout the brain". Do you mean larger TBV and differences in WM structures?

Page 9 lines 209-210. Check this sentence

Reviewer #2 (Remarks to the Author):

In this study, the authors identify potentially pathogenic variants in MAP1B in individuals with intellectual disability, decreased white matter and decreased corpus callosum volume. They first identified a truncating variant in a family with several affected individuals. By searching for additional LoF variants in >31,000 Icelanders, they identify two additional variants in families with one or more affected individuals. This is a well-written paper with convincing evidence for the role of MAP1B in mild ID. I have only minor comments.

The data regarding cognitive function is very nice. Is there additional phenotype data available? For example: Is there any information about head circumference in the carriers of MAP1B truncating variants? Do the carriers have any dysmorphic features? Do any individuals have seizures?

Are there any functional assays for HC function? Given that there is a stably expressed truncated form, would the authors consider "LoF" an appropriate label for these variants? Truncating variants may be more appropriate.

Did the authors look for rare missense variants? Given their results, would one predict that missense variants could produce the same phenotype?

Reviewer #3 (Remarks to the Author):

In this study the authors analyzed a large whole-genome sequencing dataset starting from an analysis of a pedigree with multiple individuals with ID. The overall study design is sound. The authors identified a MAP1B truncating variant (E712KfsTer10) almost completely linked to ID in the initially analyzed family, and confirmed association of MAP1B LoF variants with ID/low FSIQ by identifying two different LoF variants (E1032Ter and R1664Ter). Analysis of brain images and cognitive assessment aid delineation of phenotypes defined by MAP1B LoF variants. While the P values obtained from a simple regression analysis for ID status or IQ may not have reached to the exome-wide significance threshold (i.e. 2.5×10^{-6} , assuming that there are $\sim 20,000$ protein-coding genes), considering some supportive findings from brain imaging analysis as well as very low frequency of MAP1B truncating variants in the general population, the association between MAP1B LoF variants and ID/low IQ looks to be a likely true positive finding.

Comments:

The genome-wide simulation procedure for calculation of P value for association of ID/low IQ with a rare variant in the pedigree is unclear to me. Did the authors generated the same structure of pedigree by randomly selecting a founder from $\sim 30,000$ WGS and then calculated the highest score within each simulated pedigree and obtained P values? If my understanding is correct, independence of each simulated dataset would be a matter to be considered. In any case, including a case where my understanding is not correct, it should be recommended to provide more detailed information on the analytical procedures. It should be also encouraged to clarify what is the advantage of the method used in this study when compared with the classical and widely accepted parametric linkage analysis calculating LOD scores.

While overall association of MAP1B LoF variants with low FSIQ seems to be quite firm, there is some phenotypic variation among the mutation carriers. Specifically, there are some MAP1B LoF carriers with a normal range of FSIQ. It would be interesting to test if polygenic SNP scores for related phenotypes (e.g. IQ, educational attainment etc.) account for this observation (when such data is available). For example, if higher polygenic scores for high IQ or educational attainment were observed in individuals with a normal range of FSIQ despite of a MAP1B LoF variant, the observed result should be more reasonably interpreted.

If other clinical information of MAP1B LoF carriers (e.g. dysmorphic features including facial photographs and other comorbid symptoms such as seizures and non-neurodevelopmental phenotypes) is available, it will be greatly helpful when defining the "MAP1B deficiency syndrome".

All MAP1B LoF variants identified in this study are not located in the last exon or the last 50bp of the penultimate exon. Therefore, an analysis testing if the mutated transcripts are subjected to

nonsense-mediated decay, rather than the analysis of products from the full-length cDNA with or without the mutation, should be more meaningful.

ExAC database showed several or more LoF or likely LoF variants in MAP1B such as stop gain, frameshift, and splicing variants. These variants should be described and carefully evaluated to clearly demonstrate that the three variants in this study are somehow functionally different. Moreover, previously reported variants shown in supple table 4 should be also evaluated and discussed more in detail.

Minor points:

1. In Line 84, the authors stated that "...replicated the association with LoF variants in MAP1B". However, in a strict sense, it is not an independent replication as the authors used the same WGS dataset of ~30,000 Icelanders in the both stages of the analyses. Therefore, another word such as "further supported" or "further confirmed", would be more appropriate.

2. In Line 112, it is uncertain what "Psychopathology" stands for (diagnosis of psychiatric disorders by MINI? or personality traits such as psychopathy?).

3. Abbreviations in Table 1 (the types of cognitive test; e.g. LF, CF etc.) should be spelled out in the footnote.

Also I would like to note that this reviewer is not familiar with statistical/quantitative analysis of brain images. This part of the analysis in this manuscript would be needed to be reviewed by other researcher(s) expertized in the related field.

We would like to thank the reviewers for their time and constructive criticism. The specific comments encouraged us to reevaluate certain sections and the inclusion of ideas set forth by the reviewers, we hope, has improved the manuscript. The reviewer comments have been split up into numbered questions, with answers in bold.

Reviewers' comments:

Reviewer #1:

The authors present a WGS study identifying a frameshift variant in MAP1B that is present in 8 members of a single family 7 of whom are affected. Two other families were identified with stop-gain variants in MAP1B with carriers in both families showing patterns of intellectual impairment. Carriers of these variants showed lower white matter in MRI with these changes concentrated in the corpus callosum.

The paper is clearly written and presents an important finding that will be of interest to a diverse group of readers.

Specific comments.

1. Page 4 line 70 - Please add a sentence commenting on the 8th family member (A1?). There is some information in the supplement and I'm guess the AD is the reason for the lack of IQ data however it would be helpful to state this. For readers who only read the main text it will appear that the 8th family member is unaffected.

FAM1-A1 is deceased and so cognitive testing and MRI was not available. A sentence has been added to the main text to address this point.

2. Can you also please add a comment on individual D4 who is not a carrier but has severe ID?

We have noted the noncarrier in FAM1 and offered a possible explanation in the text.

3. **a.** Given WGS data are available for these individuals it would be very helpful to know where they sit on a polygenic risk score distribution for IQ as compared to the controls. Having information about their burden of genetic risk due to common variants would help quantify the extent to which these deficits are due solely to the MAP1B variants vs their polygenic background risk. **b.** Multiple non-carriers in this family also appear to have deficits that are presumably due to the burden of common risk variants.

We have added a subsection to the Results where we estimate the polygenic contribution, calculated from an IQ and educational attainment GWAS, in MAP1B and controls, and do not find a significant difference. Although not tested and likely underpowered, the MAP1B noncarriers do not appear to have higher polygenic score for either IQ or educational attainment either. However, assessment of this will be confounded by FAM1 being multigenerational allowing for the introduction, into the family, of a mixture of rare and common markers conferring risk of cognitive deficits.

4. Methods - Many of the analyses state that country of birth was included as a covariate. It's unclear if this is to correct for ancestry or if all participants are of Icelandic ancestry and this is to correct for a cultural effect relating to migration. Please clarify. Also if this is a correction for ancestry please elaborate on how this is a sufficient correction.

That should read county, not country, of birth. To account for possible stratification within Iceland.

5. Table 1 - Please provide the mean and se|sd on the raw scale for each test.

Table 1 has been updated with the mean and standard deviation for the unadjusted values from the cognitive tasks and IQ testing.

6. Table 2 - Please provide the se|sd for each of the imaging measures to accompany these means.

Table 2 and Supplementary Table 6 have been updated with the mean and standard deviation for the unadjusted values from the FreeSurfer MRI traits.

Minor Text suggestions

7. Page 3 line 68 - change "trunctation () associated" -> "trunctation () was associated"

We have amended this sentence.

8. Page 6 line 116 - change "and less brain-wide" -> "and lower brain-wide"

We feel less is an appropriate term to use for volume changes.

9. Page 9 lines 193-198. Consider breaking this sentence at the point where you transition from talking about the human MRI studies to the mouse knockouts.

We have split and simplified the text to present first a general background of FMR1/FMRP/FXS, followed by a paragraph about mouse and human studies, closing with a suggestion for the effect on *MAP1B* when FMRP is dysfunctional.

10. Also please check/clarify the section of this sentence talking about the mouse studies. It is unclear what is meant by "significantly larger total brain volume differences in major WM structures throughout the brain". Do you mean larger TBV and differences in WM structures?

This section has been slightly expanded and what Reviewer #1 refers to has hopefully been clarified.

11. Page 9 lines 209-210. Check this sentence

The sentence has been changed. We are emphasising the importance of *MAP1B* by linking its loss with the phenotypes observed in the *BCL11A-L* and *FMRP* syndromes.

Reviewer #2:

In this study, the authors identify potentially pathogenic variants in MAP1B in individuals with intellectual disability, decreased white matter and decreased corpus callosum volume. They first identified a truncating variant in a family with several affected individuals. By searching for additional LoF variants in >31,000 Icelanders, they identify two additional variants in families with one of more affected individuals. This is a well-written paper with convincing evidence for the role of MAP1B in mild ID.

I have only minor comments.

12. The data regarding cognitive function is very nice. Is there additional phenotype data available? For example: Is there any information about head circumference in the carriers of MAP1B truncating variants? Do the carriers have any dysmorphic features? Do any individuals have seizures?

Head circumference, height and weight were measured. However, after adjusting for age, gender and relatedness, MAP1B carriers are not significantly different compared with controls.

Dysmorphic features based on Tripi et al. (Tripi G, Roux S, Canziani T, Bonnet Brillhault F, Barthelemy C, Canziani F. Minor physical anomalies in children with autism spectrum disorder. Early Hum Dev 84, 217-223 (2008)) were evaluated by the psychologist performing the cognitive evaluation. However, no feature was observed in higher frequency in the carriers.

We have also included a short section on clinical MRI evaluation to balance the quantitative phenotyping.

FAM1-D3 has epilepsy as shown in Supplementary Table 1. We are not aware of seizure or epilepsy in any of the other carriers.

13. a. Are there any functional assays for HC function? **b.** Given that there is a stably expressed truncated form, would the authors consider "LoF" an appropriate label for these variants? Truncating variants may be more appropriate.

As we point out "in-vitro studies have shown that similarly truncated MAP1B HC can promote microtubule assembly" (Bondallaz, et al. CellMotilCytoskel, 2006. The control of microtubule stability in vitro and in transfected cells by MAP1B and SCG10). However, functional assessment of the effect of protein truncating variations in MAP1B is beyond the scope of this genetic study.

The loss-of-function terminology is applied to variants resulting in protein truncation (that includes nonsense, splice acceptor, and splice donor variants). LoF applies if the canonical function of the protein is lost or impaired. This appears to be the case for the variants described here, given the profound white matter changes and cognitive deficits. However, it is yet to be shown whether the HC truncation, loss of LC1 or both are responsible for the phenotypes observed.

14. Did the authors look for rare missense variants? Given their results, would one predict that missense variants could produce the same phenotype?

A number of missense variants were identified in the MAP1B gene. However, none were found in individuals with ID or ASD. This is in agreement with the modest missense variant constraint reported in the ExAc database for MAP1B. This has been addressed in the text.

Reviewer #3:

In this study the authors analyzed a large whole-genome sequencing dataset starting from an analysis of a pedigree with multiple individuals with ID. The overall study design is sound. The authors identified a MAP1B truncating variant (E712KfsTer10) almost completely linked to ID in the initially analyzed family, and confirmed association of MAP1B LoF variants with ID/low FSIQ by identifying two different LoF variants (E1032Ter and R1664Ter). Analysis of brain images and cognitive assessment aid delineation of phenotypes defined by MAP1B LoF variants. While the P values obtained from a simple regression analysis for ID status or IQ may not have reached to the exome-wide significance threshold (i.e. 2.5×10^{-6} , assuming that there are $\sim 20,000$ protein-coding genes), considering some supportive findings from brain imaging analysis as well as very low frequency of MAP1B truncating variants in the general population, the association between MAP1B LoF variants and ID/low IQ looks to be a likely true positive finding.

Comments:

15. The genome-wide simulation procedure for calculation of P value for association of ID/low IQ with a rare variant in the pedigree is unclear to me. Did the authors generated the same structure of pedigree by randomly selecting a founder from $\sim 30,000$ WGS and then calculated the highest score within each simulated pedigree and obtained P values? If my understanding is correct, independence of each simulated dataset would be a matter to be considered. In any case, including a case where my understanding is not correct, it should be recommended to provide more detailed information on the analytical procedures. It should be also encouraged to clarify what is the advantage of the method used in this study when compared with the classical and widely accepted parametric linkage analysis calculating LOD scores.

In order to provide more information on the analytical procedure we have added a detailed description to the methods instead of the brief summary and citing the Nature Genetics manuscript “Truncating mutations in RBM12 are associated with psychosis” which described the method originally.

Regarding LOD scores, the method section states that “Analogous to parametric linkage analysis, we assume an effect of the variant being sought, but different from parametric linkage analysis we do not search for linkage with unknown linkage variants but test particular variants having a known set of carriers.” The LOD scores capture excess of IBD sharing between individuals and ignore which variants are being shared. While this was a reasonable approach when sequencing data was not available, it is now outdated.

16. While overall association of MAP1B LoF variants with low FSIQ seems to be quite firm, there is some phenotypic variation among the mutation carriers. Specifically, there are some MAP1B LoF carriers with a normal range of FSIQ. It would be interesting to test if polygenic SNP scores for related phenotypes (e.g. IQ, educational attainment etc.) account for this observation (when such data is available). For example, if higher polygenic scores for high IQ or educational attainment were observed in individuals with a normal range of FSIQ despite of a MAP1B LoF variant, the observed result should be more reasonably interpreted.

We have added a subsection to the Results where we estimate the polygenic contribution, calculated from an IQ and educational attainment GWAS, in MAP1B and controls, and do not find a significant difference. Although not tested and likely underpowered, the MAP1B noncarriers do not appear to have higher polygenic score for either IQ or educational attainment either. However, assessment of this will be confounded by FAM1 being multigenerational allowing for the introduction, into the family, of a mixture of rare and common markers conferring risk of cognitive deficits.

17. If other clinical information of MAP1B LoF carriers (e.g. dysmorphic features including facial photographs and other comorbid symptoms such as seizures and non-neurodevelopmental phenotypes) is available, it will be greatly helpful when defining the “MAP1B deficiency syndrome”.

Head circumference, height and weight were measured. However, after adjusting for age, gender and relatedness, MAP1B carriers are not significantly different compared with controls.

Dysmorphic features based on Tripi et al. (Tripi G, Roux S, Canziani T, Bonnet Brilhault F, Barthelemy C, Canziani F. Minor physical anomalies in children with autism spectrum disorder. Early Hum Dev 84, 217-223 (2008)) were evaluated by the psychologist performing the cognitive evaluation. However, no feature was observed in higher frequency in the carriers.

We have also included a short section on clinical MRI evaluation to balance the quantitative phenotyping.

FAM1-D3 has epilepsy as shown in Supplementary Table 1. We are not aware of seizure or epilepsy in any of the other carriers.

18. All MAP1B LoF variants identified in this study are not located in the last exon or the last 50bp of the penultimate exon. Therefore, an analysis testing if the mutated transcripts are subjected to nonsense-mediated decay, rather than the analysis of products from the full-length cDNA with or without the mutation, should be more meaningful.

The three MAP1B LoF variants identified here are located in exon 5 which is 6,501 nucleotides long. In general, premature termination codons (PTCs) lead to haploinsufficiency as a result of nonsense-mediated decay (NMD) of the mutated mRNA. However, NMD efficiency is reduced when PTCs are located within large exons, are far from the next downstream exon junction complex (EJC) or are far from the normal stop codon according to Lindeboom et al. (Lindeboom RG, Supek F, Lehner B. The rules and impact of nonsense-mediated mRNA decay in human cancers. Nat Genet 48, 1112-1118 (2016)). As this applies to all of the variants described here it is unlikely that the NMD mechanism clears the mutated transcripts and therefore truncated proteins are assumed to be formed. This was demonstrated by the Western blot study.

19. ExAC database showed several or more LoF or likely LoF variants in MAP1B such as stop gain, frameshift, and splicing variants. These variants should be described and carefully evaluated to clearly demonstrate that the three variants in this study are somehow functionally different. Moreover, previously reported variants shown in supple table 4 should be also evaluated and discussed more in detail.

We agree with Reviewer #3 that it would be useful to evaluate the LoF variant carriers previously identified in available databases and publications for cognitive and MRI phenotypes. However, we do not have access to those carriers. It is possible that some of the LoF variant carriers identified in databases and other publications will demonstrate the same cognitive and neurological deficits described here.

Functional assessment of the effect of protein truncating variations in *MAP1B* is beyond the scope of this genetic study.

Minor points:

20. In Line 84, the authors stated that “...replicated the association with LoF variants in MAP1B”. However, in a strict sense, it is not an independent replication as the authors used the same WGS dataset of ~30,000 Icelanders in the both stages of the analyses. Therefore, another word such as “further supported” or “further confirmed”, would be more appropriate.

We agree with Reviewer #3 and have rephrased to “further confirmed”.

21. In Line 112, it is uncertain what “Psychopathology” stands for (diagnosis of psychiatric disorders by MINI? or personality traits such as psychopathy?).

Here we are referring to the diagnosis of psychiatric disorders covered by the MINI. The text will be changed to reflect this more clearly.

22. Abbreviations in Table 1 (the types of cognitive test; e.g. LF, CF etc.) should be spelled out in the footnote.

Abbreviations have been explained for Table 1 and Supplementary Figure 3.

23. Also I would like to note that this reviewer is not familiar with statistical/quantitative analysis of brain images. This part of the analysis in this manuscript would be needed to be reviewed by other researcher(s) expertized in the related field.

Reviewer #1 (Remarks to the Author):

The authors have addressed the comments well and I think the revised manuscript improved.

To clarify my comment on polygenic risk the authors have compared the PRS level within the family, as might be expected with this approach there are no differences seen.

The more informative approach would be to calculate the PRS of the family in conjunction with a large population sample (which the authors are known to have access to) and report the percentile in which the affected family members sit. This would better address the question of what impact the rare variation might have in conjunction with the the common.

Reviewer #3 (Remarks to the Author):

I appreciate the authors' effort to address my concerns. The authors answered many of the comments from this Reviewer. However, there would be some misunderstanding of the concerns I have raised, as follows:

Regarding the reply to Point 16:

Comparison of polygenic scores between mutation carriers and controls is meaningful when speculating if there is contribution of background genetic factors. However, what I have mainly suggested in my previous comment was a comparison between MAP1B mutation carriers with and without ID, or an analysis of relationship between the polygenic common SNP scores and IQ within the mutation carriers. While I understand that the number of individuals would be too small, the latter approach (analysis of correlation) would provide some interpretable results.

Regarding the reply to Point 18:

It is true that NMD efficiency is reduced when PTCs are located in long exons, especially when these are distant from the next downstream exon junction. However, as shown in the paper cited in their rebuttal letter (Lindeboom et al.), NMD efficiency for PTVs in long exons is not zero. Also I could not find important information of the distance between the PTVs identified in this study and the next downstream exon junction in the main manuscript. Therefore it would merit doing experiments to confirm whether they actually lead to NMD. This is within the scope describing the nature of

variants. It is definitely interesting to see NMD using RT-PCR on patients' derived lymphoblasts or even minigene assays.

The authors performed Western blot analysis of protein products from HeLa cells transfected with plasmids containing cDNA. Even if truncated proteins were clearly seen in such in vitro assay, NMD was not ruled out.

Related to this, I agree the Reviewer #2's comment that the term "LoF" would not be appropriate if the protein products are stably expressed and there is no functional assay.

Regarding the reply to Point 19:

In ExAC, I found many stop-gain and frameshift variants. This is in contrast to authors' finding that only three frameshift/stop-gain variants in 31,463 Icelanders. I assume 31463 Icelanders might contain many relatives. In my previous comment, I would like to see any difference between the three frameshift/stop-gain variants and frameshift/stop-gain variants in ExAC. Authors did not seem to seriously consider my comment due to the reason of inaccessibility to the similar cognitive and neurological deficit evaluation. However I think they should list the variants in ExAC and discuss about their nature compared to that of the three frameshift/stop-gain variants described here. Readers should have a question why MAP1B has various such frameshift/stop-gain variants in ExAC. This should be discussed and clarified.

We thank the reviewers for this follow up criticism. We hope that our answers to their questions alleviates their concerns and the changes made improve the readability of the manuscript.

Reviewers' comments:

Reviewer #1:

The authors have addressed the comments well and I think the revised manuscript improved.

1. To clarify my comment on polygenic risk the authors have compared the PRS level within the family, as might be expected with this approach there are no differences seen. The more informative approach would be to calculate the PRS of the family in conjunction with a large population sample (which the authors are known to have access to) and report the percentile in which the affected family members sit. This would better address the question of what impact the rare variation might have in conjunction with the the common.

We did in fact compare PGS (IQ and EDU) between *MAP1B* carriers and a large population sample as this sentence reflects in the "IQ and educational attainment polygenic score in *MAP1B* carriers" subsection of the Results:

"We compared the PGS-IQ and PGS-EDU in *MAP1B* carriers (n = 13) to controls (n = 2,164) and found no significant difference ($P = 0.20$ and 0.82 , respectively; Welch Two Sample t-test)....."

Furthermore, supplementary figures 4 and 5 show the carriers' PGS (IQ and EDU) are located around the mean PGS.

Reviewer #3:

I appreciate the authors' effort to address my concerns. The authors answered many of the comments from this Reviewer. However, there would be some misunderstanding of the concerns I have raised, as follows:

2. Regarding the reply to Point 16:

Comparison of polygenic scores between mutation carriers and controls is meaningful when speculating if there is contribution of background genetic factors. However, what I have mainly suggested in my previous comment was a comparison between *MAP1B* mutation carriers with and without ID, or an analysis of relationship between the polygenic common SNP scores and IQ within the mutation carriers. While I understand that the number of individuals would be too small, the latter approach (analysis of correlation) would provide some interpretable results.

As the Reviewer points out, comparing PGS-IQ within families is statistically underpowered. We believe the sentence, and supplementary figures 4 and 5, provide the reader with enough information to recognise that the *MAP1B* variants impair IQ beyond the PGS for IQ:

"We compared the PGS-IQ and PGS-EDU in *MAP1B* carriers (n = 13) to controls (n = 2,164) and found no significant difference ($P = 0.20$ and 0.82 , respectively; Welch Two Sample t-test)....."

3. Regarding the reply to Point 18:

a. It is true that NMD efficiency is reduced when PTCs are located in long exons, especially when

these are distant from the next downstream exon junction. However, as shown in the paper cited in their rebuttal letter (Lindeboom et al.), NMD efficiency for PTVs in long exons is not zero.

We don't state that NMD is impaired. That would require a more complex experiment, beyond the scope of this association study. We demonstrate that truncated protein can be produced.

b. Also I could not find important information of the distance between the PTVs identified in this study and the next downstream exon junction in the main manuscript.

The distance between the PTVs described in our manuscript and the next downstream exon junction is indicated in Supplementary Figure 1 a. For clarity, we have added the hg38 start and stop positions of *MAP1B* exon 5 to the supplementary figure text. We have also included in Supplementary figure 1 annotations (red dotted lines) for the approximate position of the ExAc variants.

c. Therefore it would merit doing experiments to confirm whether they actually lead to NMD. This is within the scope describing the nature of variants. It is definitely interesting to see NMD using RT-PCR on patients' derived lymphoblasts or even minigene assays.

The authors performed Western blot analysis of protein products from HeLa cells transfected with plasmids containing cDNA. Even if truncated proteins were clearly seen in such in vitro assay, NMD was not ruled out.

The suggested effort would be a task for a laboratory better suited for functional studies of this nature. *MAP1B* is not expressed in blood and we do not have any samples from the *MAP1B* carriers suitable for expression studies. Furthermore, RT-PCR of long transcripts is not always straight forward.

d. Related to this, I agree the Reviewer #2's comment that the term "LoF" would not be appropriate if the protein products are stably expressed and there is no functional assay.

The *MAP1B* protein is cleaved into a heavy chain and a light chain. All of the variants described in our manuscript cause truncation of *MAP1B* and a loss of the light chain. The light chain has been shown to have numerous functions. Therefore, irrespective of whether there is loss of the light chain or the heavy chain is not functioning correctly, the use of loss of function to describe the variants is appropriate, as there is loss of functional elements of the protein.

4. Regarding the reply to Point 19:

In ExAC, I found many stop-gain and frameshift variants. This is in contrast to authors' finding that only three frameshift/stop-gain variants in 31,463 Icelanders. I assume 31463 Icelanders might contain many relatives. In my previous comment, I would like to see any difference between the three frameshift/stop-gain variants and frameshift/stop-gain variants in ExAC. Authors did not seem to seriously consider my comment due to the reason of inaccessibility to the similar cognitive and neurological deficit evaluation. However I think they should list the variants in ExAC and discuss about their nature compared to that of the three frameshift/stop-gain variants described here. Readers should have a question why *MAP1B* has various such frameshift/stop-gain variants in ExAC. This should be discussed and clarified.

ExAc identifies nine loss of function variants that are observed in the canonical transcript and have a PASS filter, in the >60,000 samples. Their combined allele frequency is approximately 1 in 10,000. We provided Supplementary Tables 3 (*MAP1B* coding variants previously reported in literature), 4 (*MAP1B* coding variants reported in public databases) and 5 (*MAP1B* copy number variations reported in literature or public databases) for readers to familiarise themselves with (ST3) what

variants have been reported in whole exome and genome sequenced autism spectrum disorder, intellectual disability and developmental delay datasets, (ST4) what loss of function variants are reported in 1000 Genomes, Database of single nucleotide polymorphisms (dbSNP), Database of genomic variation and Phenotype in Humans using Ensembl Resources (DECIPHER), Exome Sequencing Project-Exome Variant Server (EVS-ESP), Exome Aggregation Consortium (ExAc) and Genome Aggregation Database (gnomAD), and (ST5) what copy number variants are reported in NCBI's VarView, DECIPHER and the only paper citing a CNV spanning *MAP1B* (Liu et al. 2015).

In relation to the above mentioned variants and regarding the comparison of variants in the ExAc database with those described in our manuscript, we do agree with the reviewer and did consider the potential strength of being able to compare phenotypes from the ExAc individuals with those we describe. However, as those data are not available to us we cannot speculate why other groups have not seen the associations we report on. The ExAc dataset is not a sample of extended families like the Icelandic sample. We demonstrate that LoF variant in the *MAP1B* gene associate with ID in a large Icelandic family and confirm the association in two additional Icelandic families. If cognitive data were available for the subjects in the ExAc dataset that sample could be used for evaluating the impact of those LoF variants in *MAP1B*. In absence of the cognitive data the sample is of limited use for validating the association we are reporting.